## RESEARCH ARTICLE

# Myocardin-related transcription factor regulates actomyosin contractility and apical junction remodeling during vertebrate neural tube closure

Keiji Itoh*, Olga Ossipova* and Sergei Y. Sokol‡

## ABSTRACT

Myocardin-related transcription factor A (Mrtfa), also known as megakaryoblastic leukemia protein (Mkl1/MAL), associates with serum response factor (Srf) to regulate transcription in response to actin dynamics; however, the functions of Mrtfa in vertebrate embryos remain largely unknown. Here, we show that Mrtfa is required for neural plate folding in early *Xenopus* embryos. Mrtfa knockdown reduced F-actin levels and inhibited apical constriction in the neural and non-neural ectoderm. By contrast, overexpression of constitutively active Mrtfa induced apical constriction in ectodermal cells via remodeling of tricellular junctions and junctional recruitment of Myosin II. We also identify potential Mrtfa target genes in embryonic ectoderm that encode actins and many regulators of actomyosin networks and junction remodeling. Our findings suggest a role for Mrtfa in the control of morphogenetic movements during neurulation. We propose that the regulation of actomyosin contractility is an essential cellular response to Mrtfa-dependent transcriptional activation.

**KEY WORDS: MrtfA, Serum response factor, Morphogenesis, Megakaryoblastic leukemia, Mkl1, F-actin, *Xenopus* embryo, Apical constriction, Neural tube closure, Tricellular junctions**

## INTRODUCTION

Early vertebrate development involves the specification of embryonic cell fates and the regulation of morphogenetic processes that place these cells and tissues into their prospective positions. Actin remodeling plays a major role in the control of cell shape and cell motility (Buracco et al., 2019; Pollard and Cooper, 2009). In addition, actin dynamics regulate gene transcription (Olson and Nordheim, 2010), contributing to the interplay between cell fate specification and the cell movements accompanying morphogenesis. Early *Xenopus* embryos serve as an excellent *in vivo* model for studying these mechanisms because they exhibit diverse cell behaviors, such as epiboly, apical constriction, tissue involution and convergent extension, that are needed for

Department of Cell, Developmental and Regenerative Biology, Icahn School of Medicine at Mount Sinai, New York, NY 10029, USA.
*These authors contributed equally to this work

‡Author for correspondence (sergei.sokol@mssm.edu)

 S.Y.S., 0000-0002-3963-9202

gastrulation and neural tube closure (Keller et al., 2003; Keller and Sutherland, 2020; Vijayraghavan and Davidson, 2017).

The myocardin-related transcription factors Mrtfa and Mrtfb, also known as megakaryoblastic leukemia 1 and 2 (Mkl1 and Mkl2), respectively, are conserved regulators of cellular actin dynamics linking actin polymerization and transcriptional control. Accumulating evidence points to Mrtfs as having important functions in normal development and disease. In mice, Mrtfs are necessary for development of smooth muscle in the heart (Cenik et al., 2016; Li et al., 2005; Mokalled et al., 2015). *Mrtfa*$^{-/-}$ mice are viable (Li et al., 2006; Sun et al., 2006b), whereas *Mrtfb*$^{-/-}$ mice die between E13.5 and E15.5 due to cardiovascular defects (Li et al., 2005; Oh et al., 2005). In humans, Mrtfa is primarily associated with acute megakaryoblastic leukemia in children (Mercher et al., 2001).

Mrtfs act as transcriptional co-factors by forming a complex with serum response factor (Srf), an evolutionary conserved protein (Cen et al., 2003; Miralles et al., 2003; Olson and Nordheim, 2010; Wang et al., 2002). Mrtfs contain a transcriptional activation domain, whereas Srf has a conserved DNA-binding domain. The binding of the Mrtf/Srf complex to CArG motifs at target gene promoters leads to transcriptional activation. Unsurprisingly, *Srf*$^{-/-}$ mice die due to gastrulation and mesoderm specification defects (Arsenian et al., 1998; Niu et al., 2005). Conditional Srf knockout mice exhibit abnormal angiogenesis, neural, cardiac and craniofacial development (Franco et al., 2008; Niu et al., 2005; Stritt et al., 2009; Vasudevan and Soriano, 2014). The formation of the active Srf/Mrtf transcriptional complex in the nucleus is inhibited by monomeric G-actin that sequesters Mrtfs in the cytoplasm (Cen et al., 2003; Miralles et al., 2003). The removal of the G-actin-binding domain can produce a constitutively active form of Mrtf (Geneste et al., 2002; Miralles et al., 2003; Weissbach et al., 2016). Consistent with this mechanism, F-actin polymerization reduces G-actin levels and promotes Mrtf nuclear localization and transcriptional activation of target genes (Esnault et al., 2014; Medjkane et al., 2009). Supporting this model, Mrtf and Srf regulate motility of various cell lines (Kim et al., 2017; Liao et al., 2014; Medjkane et al., 2009; Morita et al., 2007; Seifert and Posern, 2017), and border cell migration in *Drosophila* (Salvany et al., 2014; Somogyi and Rørth, 2004).

In this study, we investigate the roles of Mrtfa during *Xenopus* embryonic development. We show that Mrtfa knockdown reduced F-actin levels and inhibited apical constriction in the neuroectoderm, whereas overexpression of constitutively active Mrtf triggered apical constriction in non-neural ectoderm via remodeling of tricellular junctions and Myosin II activation. We also find that putative Mrtfa target genes in early embryos are associated with actomyosin cytoskeleton and cell junction remodeling. Our observations indicate that Mrtfa regulates actomyosin contractility during vertebrate neural tube closure.

**DEVELOPMENT**

## RESULTS

### Mrtfa acts as a transcriptional activator in early *Xenopus* embryos

Previous studies reported that Mrtfs stimulate target gene expression after forming a complex with Srf needed for DNA binding (Cen et al., 2003; Miralles et al., 2003; Olson and Nordheim, 2010; Wang et al., 2002). To confirm the function of Mrtfa in early *Xenopus* embryos, we deleted several functional domains of Mrtf (Fig. 1A). Constitutively active Mrtfa (ΔN-Mrtf) lacks the G-actin-binding N-terminal motifs that serve a negative regulatory role (Miralles et al., 2003; Vartiainen et al., 2007). We also generated Mrtfa lacking the C-terminal transcriptional activation domain, because such constructs were reported to inhibit MRTF function (ΔC-Mrtf) (Cen et al., 2003; Miralles et al., 2003; Zaromytidou et al., 2006). Since the association of Mrtf with SRF is crucial for DNA binding and transcriptional activation, we made a deletion construct that does not bind to SRF (ΔB-Mrtf) (Zaromytidou et al., 2006). We argue that the further analysis of these constructs should provide mechanistic insights for developmental functions of Mrtfa.

RNAs encoding the truncated Mrtfa proteins were injected into *Xenopus* early embryos to examine protein expression levels and assess the effects of various Mrtfa constructs on the transcriptional activation of *p3DA-Luc*, a Mrtf/Srf-specific luciferase reporter containing three Srf-binding sites (Busche et al., 2008; Hill et al., 1995). The generated constructs were well expressed in early embryos (Fig. S1). Consistent with the inhibitory role of G-actin, ΔN-Mrtf activated *p3DA-Luc* more strongly than wild-type Mrtfa (Fig. 1B). The ΔC-Mrtf and ΔB-Mrtf constructs did not activate the reporter but rather exhibited a dominant interfering activity (Fig. 1C,D). These observations confirm that Mrtfa retained its conserved transcriptional activation function in early *Xenopus* development.

### Mrtfa is necessary for *Xenopus* neural tube closure

For visualizing spatial patterns of Mrtfa expression during neurulation, we performed whole-mount *in situ* hybridization. Compared with *Sox2*, a neural plate marker, *Mrtfa* is more broadly expressed in dorsal ectoderm at neurula stages (Fig. S2). Transcripts for *SRF*, the co-factor for Mrtfa, were also present in the neural plate, in addition to the strong expression in paraxial mesoderm (Fig. S2).

To study a role of Mrtfs in early neural development, we designed splicing-blocking MOs that are specific for the *mrtfa* gene. The successful targeting of the junction sequences between the second intron and the third exon has been validated by RT-PCR (Fig. 2A). Unilateral injections of *mrtfa* MO, but not the control MO, caused defects in neural tube closure (Fig. 2B-D,F). Injection of 100-200 pg of Mrtfa RNA partly rescued the observed abnormalities confirming specificity (Fig. 2D-F). These observations establish a requirement for Mrtfa in morphogenetic processes that accompany neural tube closure.

To define the observed knockdown phenotype at the level of cell morphology, Mrtfa MO was co-injected with RNA encoding membrane-associated GFP (MyrGFP) and we measured apical domain size and the levels of junctional F-actin after phalloidin staining.

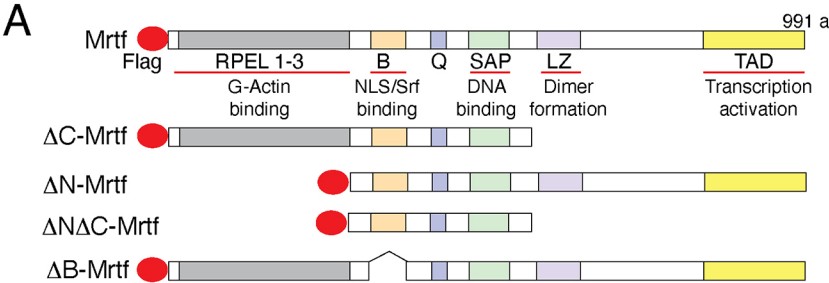

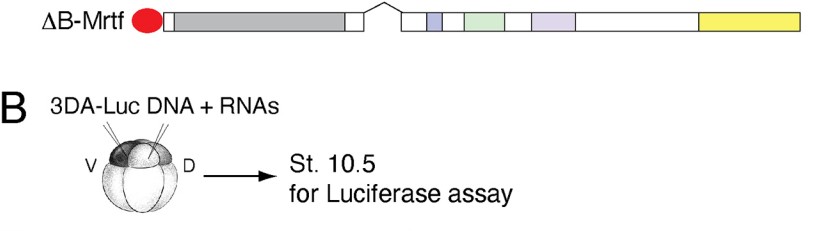

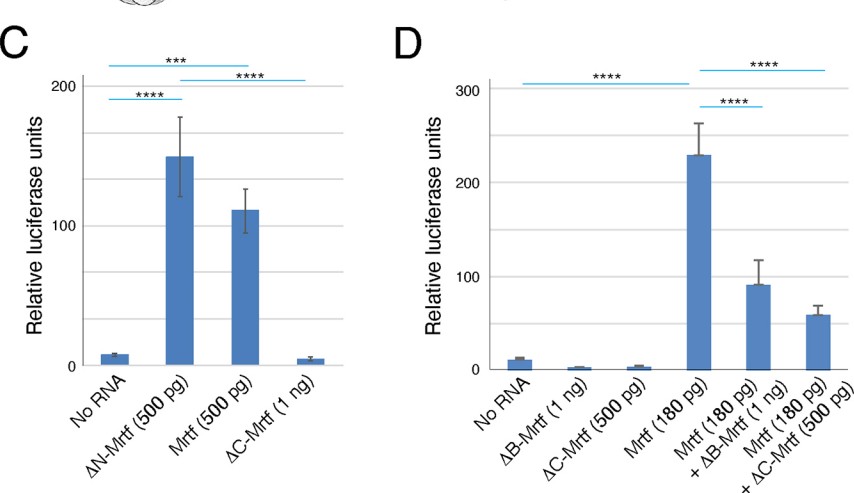

**Fig. 1. Distinct effects of Mrtfa constructs on Srf reporter activation.** (A) Domain structure of Mrtfa indicating G-actin binding, nuclear localization signal (NLS)/serum response factor (Srf) binding, DNA binding, dimer formation and transcriptional activation domains. (B) Experimental scheme for transient *3DA-Luc* reporter assays in *Xenopus* ectoderm. (C) Transcriptional activation of *3DA-Luc* DNA by wild-type Mrtfa, ΔC-Mrtfa and ΔN-Mrtfa constructs. (D) ΔC-Mrtfa and ΔB-Mrtf inhibit endogenous and exogenous Mrtf activity. Data are mean±s.d. and are representative of three to five independent experiments. One-way ANOVA Bonferroni's multiple comparison test: ***$P<0.001$, ****$P<0.0001$.

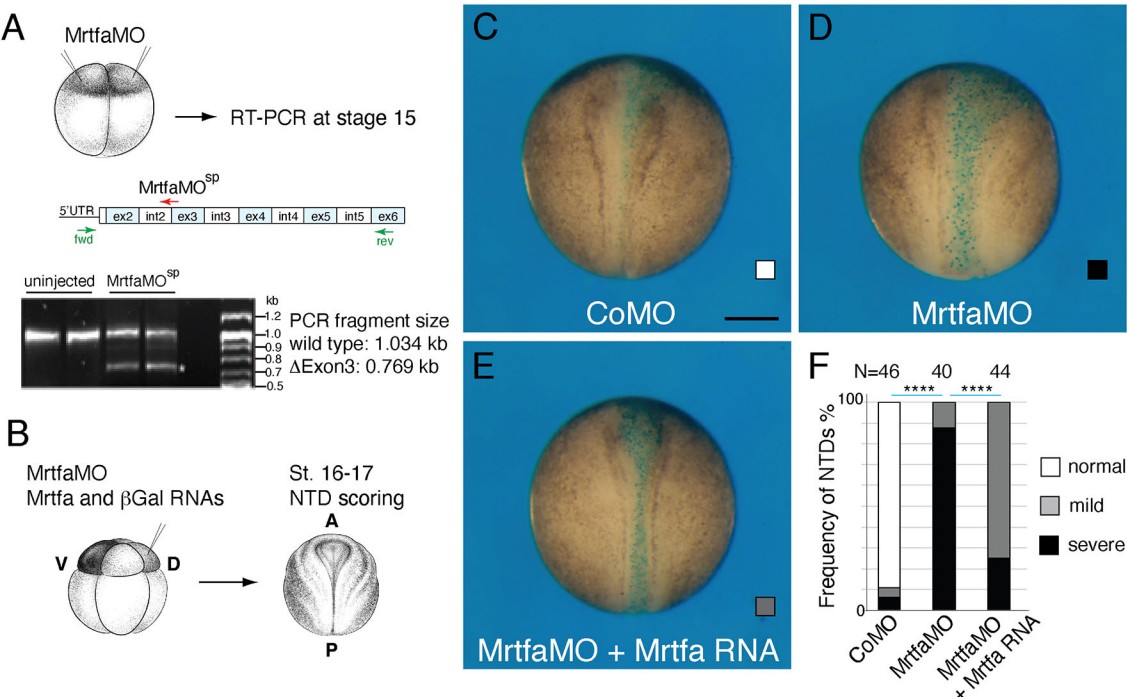

**Fig. 2. Mrtfa is required for _Xenopus_ neurulation.** (A) Validation of _Mrtfa_ knockdown. Experimental scheme. Embryos were injected animally with Mrtfa-MO^sp (25 ng). Mrtfa gene structure with indicated exon (ex) and intron (int) boundaries is shown above the gel. Mrtfa-MO^sp targets the junction of intron 2 and exon 3. RT-PCR with specific primers shows the removal of exon 3 in the morphants. The MO effect was analyzed in duplicate samples as indicated. The results are representative of two different experiments. (B-F) Neurulation defects in Mrtf morphants. (B) One dorso-animal site of eight-cell embryos was injected with control MO, with Mrtfa-MO^sp (40 ng each) or with Mrtfa-MO^sp, and with Mrtfa RNA (100 pg) and β-Gal RNA (40 pg), as indicated. Neural tube closure defects were assessed when control embryos reached stage 16-17. (C-E) Dorsal view of representative embryos, anterior is at the top. Scale bar: 300 μm. (F) Quantification of the experiments shown in C-E. Neural tube closure defects were scored as severe, mild or normal. Phenotype frequencies and the total numbers of scored embryos are indicated for each group. Data are representative of three independent experiments. Chi-square test: ****_P_<0.0001.

Morphant cells had an expanded apical domain and significantly reduced phalloidin staining intensity (Fig. 3A-E). In live-imaging studies, we also observed the increased apical domain size and decreased F-actin levels in Mrtfa-depleted ectoderm when compared to the control tissue (Movie 1). We conclude that Mrtfa is necessary for apical constriction and junctional dynamics during neural tube closure.

### Mrtfa induces apical constriction in nonneural ectoderm

Our loss-of-function studies indicate that Mrtfa is involved in actomyosin contractility regulation. For gain-of function studies, we used non-neural ectoderm because of its low level of Myosin II activity. We compared the effects of Mrtfa constructs with enhanced or inhibited transcriptional activity on ectoderm apical domain size. Stage 13 ectoderm cells expressing ΔN-Mrtf RNA exhibited progressive reduction of the apical domain when compared to the control cells expressing only the lineage tracer (Myr-GFP) (Fig. 4A-C,E). Cryosection analysis confirmed that the decrease of apical domain size was accompanied with the expansion of the basolateral domain (Fig. S3A-D). These results indicate that active Mrtf promotes ectopic apical constriction during gastrulation. To confirm that ΔN-Mrtf induces apical constriction, time-lapse imaging was carried out between stages 10.5 and 12.5, and apical constriction was observed in 46% of ΔN-Mrtf-expressing cells (_n_=143). By contrast, the reduction of the apical domain was found in fewer than 4% of control cells lacking ΔN-Mrtf (_n_=120). Notably, ΔN-Mrtf did not cause the hyperpigmentation that marks the activity of other apical constriction inducers, such as Plekhg5, Shroom3 or LMO7 (Haigo et al., 2003; Matsuda et al., 2022; Popov et al., 2018).

We next evaluated whether the induction of apical constriction by ΔN-Mrtf requires its transcriptional activity. We therefore tested ΔNΔC-Mrtf, which lacks transcription activation domain and is expressed at similar levels in ectoderm cells (Fig. S1B). ΔNΔC-Mrtf was unable to trigger apical constriction (Fig. 4D-E). These results suggest that transcriptional activation is crucial for the induction of apical constriction by Mrtf. Reinforcing this conclusion, the reduction of apical domain size in response to ΔN-Mrtf has been rescued by ΔC-Mrtf, a dominant interfering construct (Fig. S4A-F). Furthermore, ΔB-Mrtf, a construct lacking the SRF-binding domain and unable to activate transcription, lost the ability to induce apical constriction (Fig. S5A-E). These observations argue that the effect of Mrtfa on apical constriction is due to its transcriptional activity.

### Mrtfa controls Myosin II activity and TCJ remodeling

To address whether Mrtfa alters cell shape, we carried out live imaging of embryonic ectoderm mosaically expressing ΔN-Mrtf and compared them to control cells in stage 11-12 embryos. We noticed that ΔN-Mrtf-expressing ectoderm cells had enhanced F-actin dynamics (F-actin flashes and membrane-associated puncta) and acquired the convex (round) shape rather than the hexagonal shape of their neighbors (Movie 2). A similar cell shape change associated with reduced apical domain size was observed in ectoderm at stage 13 (Fig. 4A-E). This change likely reflects altered cortical tension or cell adhesion (Maitre et al., 2012; Sedzinski et al., 2016; Winklbauer, 2015), consistent with the regulation of the actomyosin cytoskeleton by Mrtfa.

To gain additional mechanistic insights, we assessed the composition of the tricellular junctions (TCJs). ΔN-Mrtf selectively

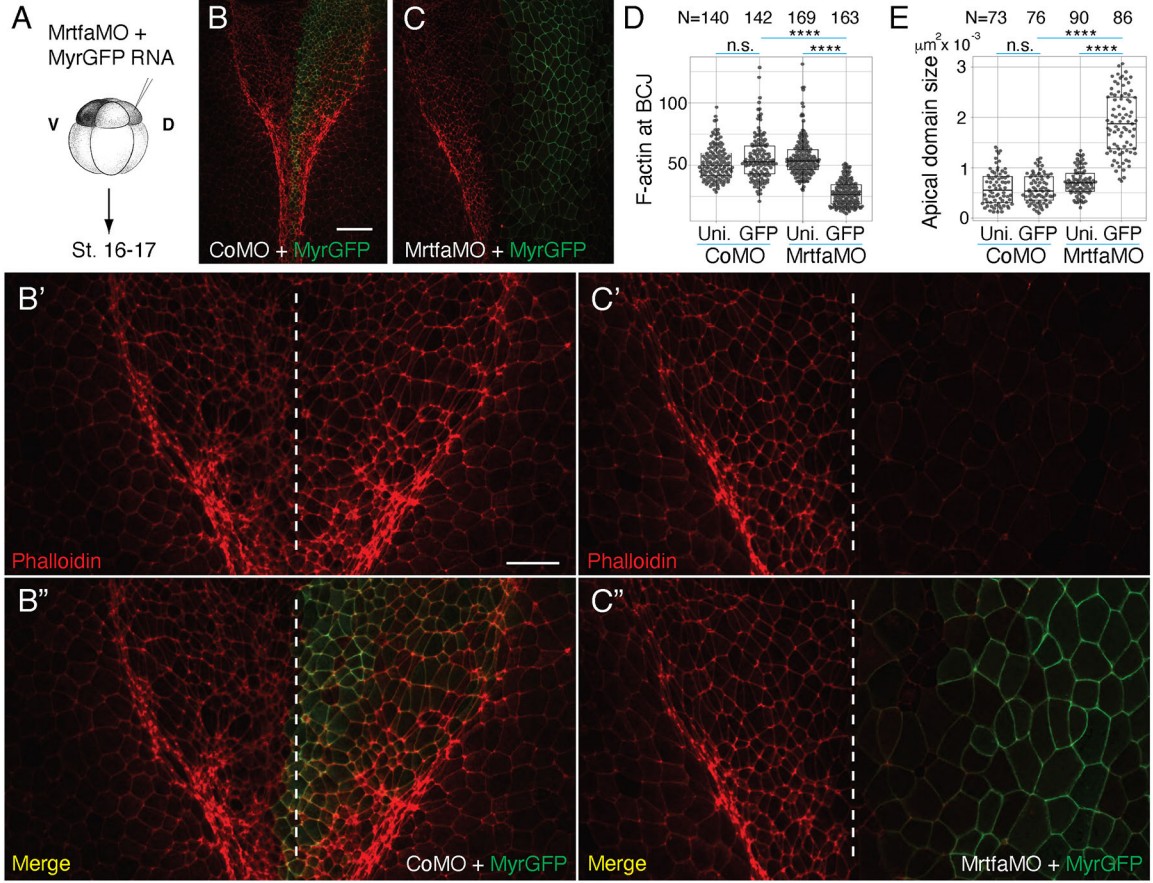

**Fig. 3. Mrtfa depletion leads to reduced F-actin levels and increases apical domain size in the neural plate.** (A) One dorso-animal site of eight-cell embryos was injected with control MO (Co MO) or Mrtfa-MOᵜ (40 ng each) with lineage tracer MyrGFP (10 pg) and allowed to develop until control embryos reached stage 16-17. (B-E) Neural plate was dissected from the injected embryos that were stained with phalloidin. Neural plate images from the CoMO- (B-B″) or Mrtf-MOᵜ (C-C″) -injected embryos at low (B,C) or high (B′,B″,C′,C″) magnification at the level of hindbrain. Midline is indicated with a white dashed line (B′,B″,C′,C″). Scale bars: 100 μm for B,C; 50 μm for B′,B″,C′,C″. (D) Quantification of F-actin intensity at the bicellular junctions (BCJs). Injected (GFP) and uninjected (Uni.) cells were scored in the neural plate at the hindbrain region, as indicated. N=numbers of junctions scored. (E) Quantification of apical domain size. N=numbers of cells scored. Kruskal–Wallis test: ****$P<0.0001$; n.s., not significant. The box plots indicate the median and quartiles.

reduced the distribution of Tricellulin (Higashi and Miller, 2017) at TCJs and increased it at the bicellular junctions (BCJs) (Fig. 5A-F,H). Similar changes were observed for F-actin (Fig. 5A-E,G,I). Notably, fluorescence of both Tricellulin and F-actin spread from TCJs to BCJs, consistent with enhanced junctional remodeling (Fig. 5F-I). Although the change in Tricellulin and F-actin localization was observed in both uniformly and mosaically expressing cells (Fig. 5F-I), mosaic ΔN-Mrtf-expressing cells were more prone to changing shape when they were next to wild-type rather than ΔN-Mrtf-expressing neighbors.

We next evaluated the localization of Myosin II using the recombinant antibody Sf9 (Vielemeyer et al., 2010). We found that Myosin II was enriched at the periphery of cells that express ΔN-Mrtf either uniformly or mosaically (Fig. 6A-F) (Movie 2). Such enhancement was not observed for F-actin (Fig. 6G,H), in parallel with our observations that Sf9 does not fully overlap with membrane-associated F-actin puncta (Movie 2). The intensities of Myosin II and F-actin at the medial apical cortex did not increase but instead decreased in ΔN-Mrtf-expressing cells (Fig. 6A-H), which is different from the increase in medioapical F-actin with apical constriction inducers such as Shroom3, GEF-H1, Lmo7 and Plekhg5 (Haigo et al., 2003; Itoh et al., 2014; Popov et al., 2018; Matsuda et al., 2022). Levels of Sf9 in whole cells were equivalent in ΔN-Mrtf expressing and control cells (Fig. 6I,J), indicating that

the localization of Myosin II to the junctions is specifically promoted by ΔN-Mrtf. Thus, the more pronounced cell shape change in mosaic cells is likely due to mismatched cortical tension at the junction between ΔN-Mrtf-expressing and control cells.

We next studied the effect of the Myosin II phosphatase Mypt1. Mypt1 counteracted the Sf9 accumulation induced by ΔN-Mrtf (Fig. 7A-E) and increased apical domain size (Fig. 7F), consistent with the ΔN-Mrtf-dependent regulation of Myosin II activity. Together, these findings suggest that Mrtfa triggers actomyosin contractility by enhancing the presence of Myosin II, Tricellulin and F-actin at cellular junctions.

### Transcriptome analysis identifies putative Mrtfa targets

To identify putative Mrtf gene targets, we carried out RNA sequencing of ectoderm cells expressing ΔN-Mrtf. The results show that the top Mrtf-induced genes encode various actins, and actin- and Myosin II-associated proteins (Fig. 8A,B). Consistent with this conclusion, gene ontology analysis (biological process) indicates the involvement of putative Mrtf targets in mesoderm and muscle development, in mesenchymal cell migration, in actin cytoskeleton organization, and in morphogenesis (Fig. 8C). To confirm the results of the RNA sequencing, several of the targets, including *acta2*, *actc1* and *myl3* were validated by quantitative RT-PCR

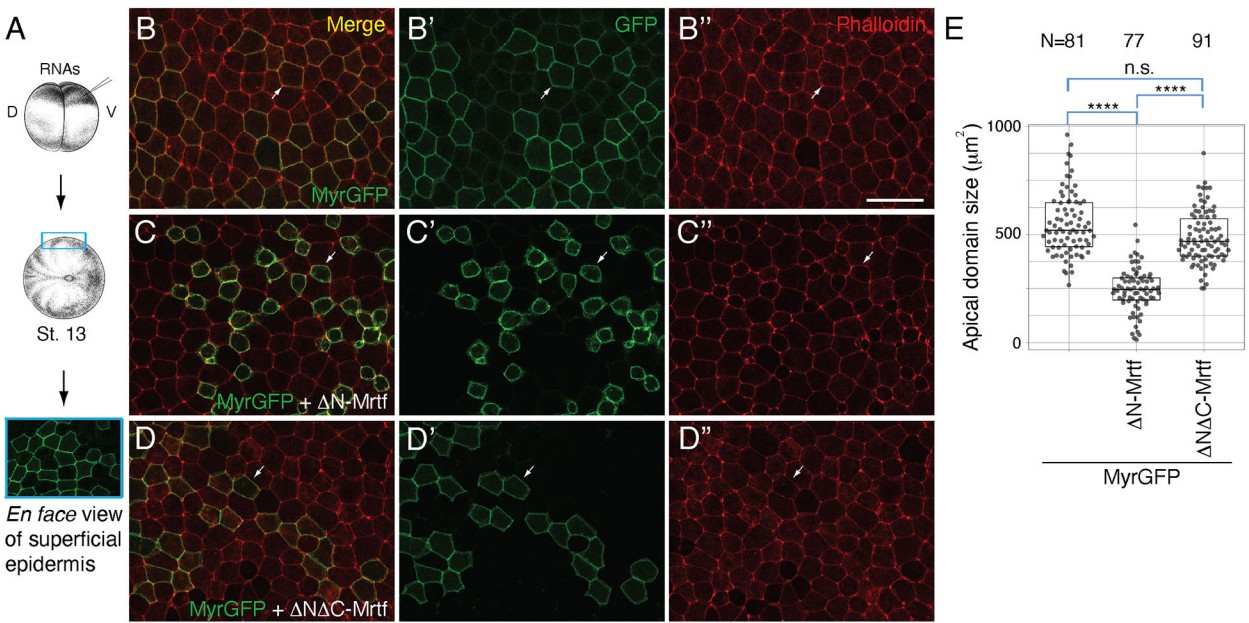

**Fig. 4. Constitutively active Mrtf induces apical constriction in non-neural ectoderm.** (A) Experimental scheme. (B-D″) Four-cell embryos were injected into one ventral-animal blastomere with MyrGFP RNA (25 pg) either alone (B-B″) or with 250 pg ΔN-Mrtf RNA (C-C″) or ΔNΔC-Mrtf RNA (D-D″). Superficial epidermal ectoderm of the injected embryos at stage 13 was imaged after phalloidin staining (B-D). Arrows indicate the cells mosaically expressing GFP. Scale bar: 50 μm. (E) Apical domain size of mosaic cells was measured. N=numbers of cells scored. Kruskal–Wallis test: ****P<0.0001; n.s., not significant. The box plot indicates the median and quartiles.

(Fig. 8D). These findings indicate that *in vivo* Mrtf targets in *Xenopus* early embryos closely match the gene targets previously identified *in vitro* in mammalian cultured cells (Esnault et al., 2014; Medjkane et al., 2009; Selvaraj and Prywes, 2004; Sun et al., 2006a).

We conclude that the actomyosin network proteins are the predominant cellular targets of Mrtf during neural tube closure. More-detailed analysis of the target genes would be necessary to elucidate the full complexity of the Mrtfa-dependent gene regulatory network.

## DISCUSSION

Our loss- and gain-of-function experiments document a role of Mrtfa in regulating actomyosin contractility during *Xenopus* neural tube closure. Embryos with reduced Mrtfa levels have neural tube closure defects and exhibit enlarged apical domain, suggesting that Mrtfa is required for apical constriction. Notably, these defects have not been described in mice lacking *Mrtfa* or *Mrtfb* genes, likely due to the functional redundancy. By contrast, we observed that a constitutively active Mrtf induced apical constriction in non-neural ectoderm. This effect required transcriptional activation because both ΔNΔC-Mrtfa, a construct with a deleted transcriptional activation domain, and ΔB-Mrtf, which does not interact with SRF, lacked the ability to induce apical constriction.

The effects of Mrtfa loss of function on F-actin levels is likely due to the lack of activation of several actin genes. In addition, the active Mtrfa construct may induce apical constriction by affecting Myosin II and F-actin. Consistent with enhanced actomyosin contractility, we observe junctional enrichment of the Myosin II heavy chain that is evident by the accumulation of SF9 fluorescence in BCJs of ΔN-Mrtfa-containing ectoderm cells. Before their apical domain size is reduced, ectoderm cells expressing ΔN-Mrtfa exhibit relocalization of Tricellulin and F-actin from TCJs to BCJs. At BCJs, F-actin and differentially enriched Myosin II in ΔN-Mrtf expressing cells can

promote actomyosin contraction. We propose that the mismatched cortical tension at the junction between ΔN-Mrtf-expressing cells and control cells promotes convex cell shape, ultimately resulting in apical constriction. Increased cell tension and apical domain reduction were previously associated with TCJ remodeling and rounded cell shape (Arnold et al., 2019; Bosveld and Bellaïche, 2020; Cho et al., 2022; Higashi and Miller, 2017). Of note, apical domain reduction triggered by ΔN-Mrtf is observed in the absence of medioapical actomyosin enrichment and hyperpigmentation, as reported for apical constriction induced by Shroom3, Plekhg5 or Lmo7 (Haigo et al., 2003; Matsuda et al., 2022; Popov et al., 2018). The hyperpigmentation probably reflects melanosome maturation and redistribution to the apical cortex (Fairbank et al., 2006). These differences point to a distinct apical constriction mechanism underlying Mrtfa effects.

Neural tube closure requires many regulators of apical constriction and junction remodeling, including Shroom3, N-cadherin and Myosin II, as well as planar cell polarity genes (Nandadasa et al., 2009; Nishimura et al., 2012; Baldwin et al., 2022; Suzuki et al., 2012; Vijayraghavan and Davidson, 2017). Our study highlights a previously unreported role of Mrtfa in linking apical constriction and junction remodeling during neural tube closure. Mrtfa may act by inducing several actomyosin-associated genes, including *myl-3* (encoding a Myosin II light chain), and appears to control the junctional accumulation of Myosin II. Besides the genes directly linked to actomyosin networks, we observed strong upregulation of many other genes, such as *dennd2a* (encoding a regulator of vesicular trafficking; Yoshimura et al., 2010), or *pacsin3* (encoding a signaling protein; Dumont and Lehtonen, 2022), suggesting that they may also participate in the control of cell behaviors by Mrtfa during morphogenesis. Detailed follow-up analysis of the differentially expressed genes identified by RNA sequencing should provide a better understanding of how Mrtfa stabilizes and activates actomyosin contractility, and the resulting changes in cell morphology.

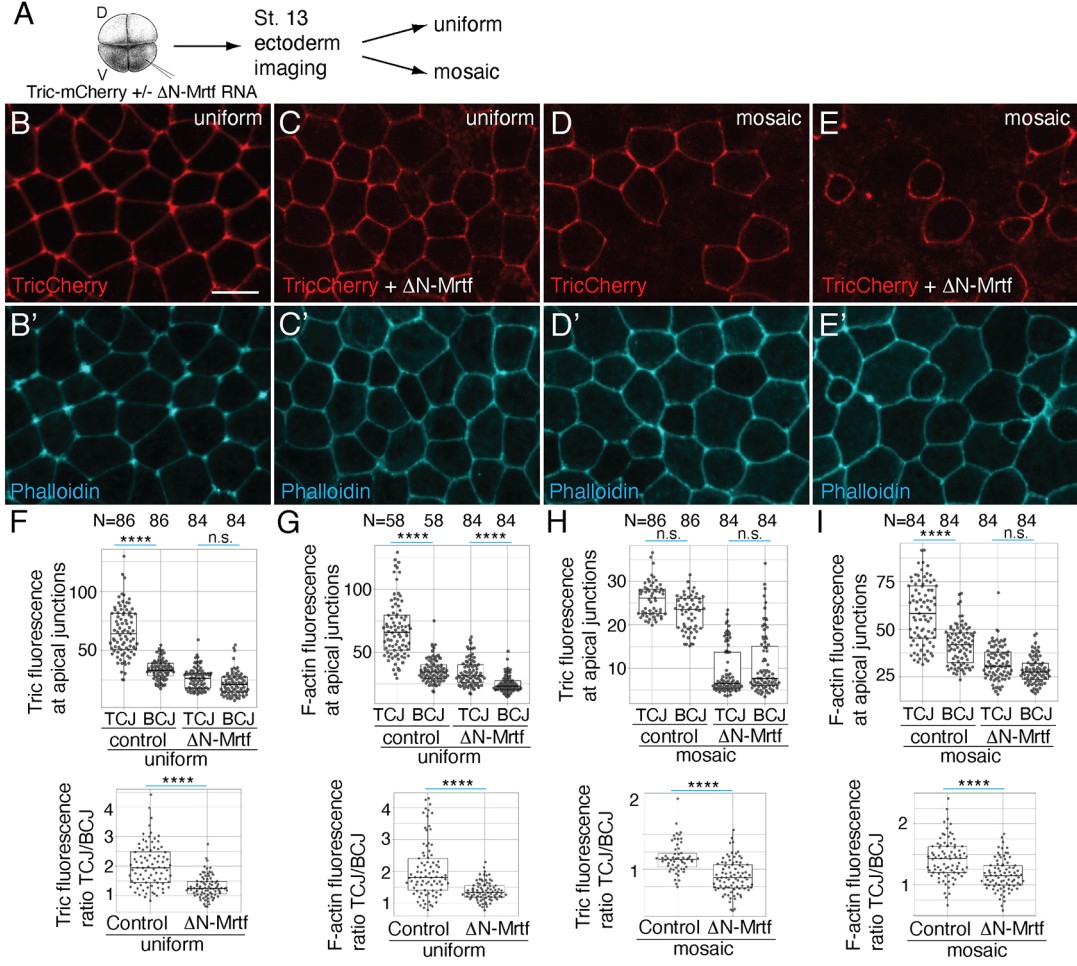

**Fig. 5. Effects of ΔN-Mrtf on Tricellulin and F-actin at the tricellular junctions.** (A) Experimental scheme. (B-E′) Four-cell embryos were injected into one ventral-animal blastomere with 25 pg Tricellulin-mCherry (TricCherry) RNA alone (B,B′,D,D′) or with 250 pg ΔN-Mrtf RNA (C,C′,E,E′). Stage 13 embryos were fixed and stained with phalloidin. (B-E) Representative ectoderm images with uniform (B,C) or mosaic (D,E) TricCherry expression; (B′-E′) phalloidin staining. Scale bar: 20 µm. (F-I) Quantification of TricCherry (F,H) or F-actin (G,I) fluorescence intensity at tricellular (TCJs) and bicellular (BCJs) junctions (top row). Quantification was carried out for cells with uniform (F,G) or mosaic (H,I) expression of Tric-Cherry alone (control) or with ΔN-Mrtf. Intensities of Tric and F-actin were measured at the junction between a mosaic and wild-type cell. Bottom row: ratios of fluorescence intensity at TCJs over BCJs. N=numbers of junctions scored. The Kruskal–Wallis test was used for the top four graphs; the Mann–Whitney test was used for the bottom four graphs: ****$P<0.0001$; n.s., not significant. The box plots indicate the median and quartiles.

Because of their binding to G-actin, Mrtfs are believed to be responsive to tensile forces. Several reports have demonstrated that Mrtfs are affected by force-dependent actin dynamics *in vitro* (Olson and Nordheim, 2010; Trembley et al., 2018; Zhao et al., 2007).

F-actin polymerization that depends on Mrtfa is expected to reduce G-actin inhibition, leading to upregulated Mrtfa activity and increased actin transcription. Whereas the nuclear localization of the fly Mrtf homolog is required for border cell migration *in vivo* (Somogyi and Rørth, 2004), little is known about Mrtfa subcellular localization in vertebrate embryos. One current limitation of our work is lack of specific antibodies required to examine endogenous Mrtfa temporal and spatial regulation during morphogenesis. Such studies are warranted to link transcriptional regulation of Mrtfa target genes with corresponding cell behaviors during morphogenesis.

## MATERIALS AND METHODS
### Plasmids, *in vitro* RNA synthesis and morpholino oligonucleotides
pCS2-Flag-Mrtf plasmids have been generated by PCR from the *X. laevis* cDNA clone for *mrtfa.S* (Accession Number NM_001088851) obtained

from Horizon. ΔN-Mrtf (lacking amino acids 6-211), ΔC-Mrtf (lacking amino acids 594-991), ΔB-Mrtf (lacking amino acids 323-367) were produced using single primer mutagenesis with specific primers (Makarova et al., 2000). ΔNΔC-Mrtf was generated from ΔN-Mrtf. All constructs have been confirmed by sequencing. pCS2-hMypt-T696A, pCS2-FlagGFP and pCS2-Scarlet-UtrCH have been described previously (Itoh et al., 2014; Matsuda and Sokol, 2025). mNeonGreen-Sf9 (Vielemeyer et al., 2010) and Tricellulin-mCherry (Higashi et al., 2016) were gifts from Ann Miller (University of Michigan, Ann Arbor, MI, USA).

Capped mRNAs were synthesized using the Ambion mMessage mMachine kit (ThermoFisher). Linearized RNA templates were made from pCS2-FlagMrtf derivatives, pCS2-Scarlet-UtrCH, pCS2-myr-tagGFP, pCS2-myr-tagRFP, pCS2-nucβGal, pCS2-Tricellulin-mCherry, pCS2-Sf9-mNeonGreen, pCS2-hMypt-T696A and pCS2-FlagGFP.

Splicing-blocking morpholinos to Mrtfa (Mrtfa MO[sp]) and control MO were purchased from Gene Tools. The MOs had the following sequences: Mrtfa MO[sp], 5′-GGTGACTGGGACCTGAAACAGAAAT-3′; control MO, 5′-CCTCTTACCTCAGTTACAATTTATA-3′.

### *Xenopus* embryo culture, microinjections and animal cap assay
Wild-type *Xenopus laevis* were maintained following the Guide for the Care and Use of Laboratory Animals of the National Institutes of Health. A

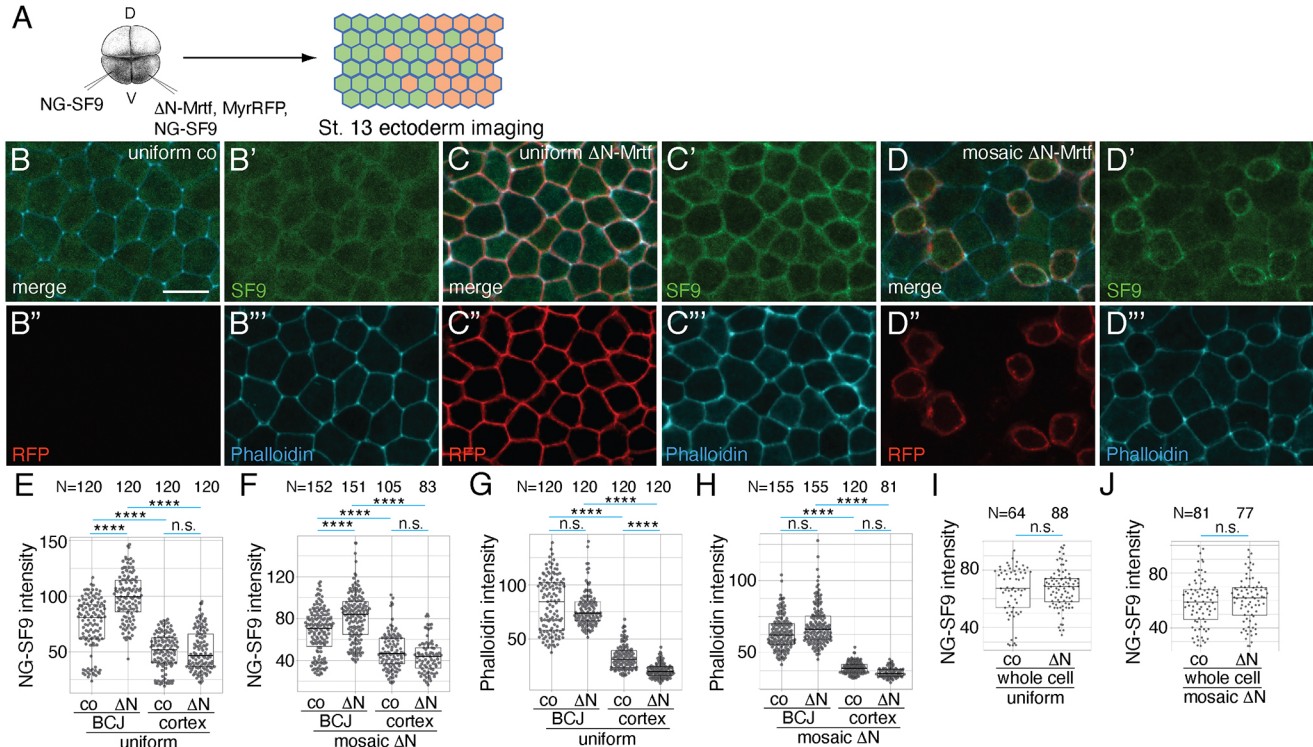

**Fig. 6. ΔN-Mrtf promotes the junctional accumulation of Myosin II.** (A) Experimental scheme. Four-cell embryos were injected into one or the other ventral-animal blastomere with 25 pg mNeonGreen-SF9 (NG-SF9), 250 pg ΔN-Mrtf or 20 pg MyrRFP RNAs, as indicated. Injected embryos were fixed at stage 13, stained with phalloidin and imaged. (B-B‴) Ectoderm cells uniformly expressing NG-SF9 without MyrRFP (uniform co). (C-C‴) Ectoderm cells uniformly expressing NG-SF9 and ΔN-Mrtf labelled with MyrRFP. (D-D‴) Cells mosaically expressing ΔN-Mrtf and NG-SF9 are labelled with MyrRFP, surrounding cells express only NG-SF9. (E-H) Quantification of fluorescence intensity of NG-SF9 (E,F) and phalloidin (G,H) at bicellular junctions (BCJs) and medial apical cortex (cortex); control cells expressing NG-SF9 only (co) are compared with cells expressing ΔN-Mrtf, MyrRFP and NG-SF9 (ΔN). Intensities of NG-SF9 and phalloidin were measured at the junction between the mosaic ΔN-Mrtf-expressing cell and control cell (ΔN), or at the junction between control cells (co). (I,J) Quantification of total NG-SF9 levels (whole cell). N=numbers of cells or junctions measured. Scale bar: 20 μm. The Kruskal–Wallis test was used for E-H; the Mann–Whitney test was used for I,J: ****$P$<0.0001; n.s., not significant. The box plots indicate the median and quartiles.

protocol for animal use was approved by the Institutional Animal Care and Use Committee (IACUC) of the Icahn School of Medicine at Mount Sinai. *In vitro* fertilization and culture of *Xenopus laevis* embryos were carried out as previously described (Itoh et al., 2005). Staging was according to (Nieuwkoop and Faber, 1967). For microinjections, 2- to 8-cell embryos were transferred into 3% Ficoll400 (Sigma) in 0.5× Marc's Modified Ringer's (MMR) solution (50 mM NaCl, 1 mM KCl, 1 mM CaCl$_2$, 0.5 mM MgCl$_2$ and 2.5 mM HEPES at pH 7.4) (Peng, 1991) and 10 nl of *3DA-Luc* DNA (a gift from G. Posern, Martin Luther University Halle-Wittenberg, Germany), mRNA or MO solution were injected into one, two or four blastomeres of 2- to 8-cell embryos, either animally or sub-equatorially. Injected embryos were transferred into 0.1× MMR at blastula stages. Amounts of injected mRNA per embryo have been optimized in preliminary dose-response experiments and are indicated in figure legends. For lineage tracing, β-galactosidase staining was carried out with X-Gal as described previously (Itoh et al., 2014). Embryo morphology was imaged using Leica M12 stereomicroscope.

### RNA sequencing and RT-PCR

For RT-PCR, two cell embryos were injected with Mrtfa-MO$^{sp}$ and cultured until stage 15. RNA was extracted from five embryos, using the RNeasy kit (Qiagen). For RT-PCR, cDNA was made from 1-2 μg of total RNA using the first-strand cDNA synthesis kit (Invitrogen) according to the manufacturer's instructions. Primers used for RT-PCR are listed in Table S1.

For RNA sequencing, Flag-ΔN-Mrtf (0.5 ng) RNA was injected twice into the animal pole region of two cell embryos. Ectoderm explants were prepared from injected and control uninjected embryos when siblings reached stage 9-9.5 and cultured until stage 11. RNA was extracted from 30 ectoderm

explants at stage 11 using RNeasy kit (Qiagen). cDNA library preparation, paired-end 150 bp sequencing using Illumina HiSeq2000 analyzers and bioinformatics analysis were performed by Novogene. The raw reads were filtered to remove reads containing adapters and of low quality. RNA counts were normalized to an uninjected control set. Sequences were mapped to the *Xenopus* genome version XL-9.1_v1.8.3.2 at http://www.xenbase.org/other/static/ftpDatafiles.jsp using Hisat2. The differentially expressed genes (DEGs) were detected using DESeq (Anders and Huber, 2010) with a twofold change cutoff. The *P*-value estimation was based on the negative binomial distribution, using the Benjamini-Hochberg estimation model with the adjusted *P*<0.05. RNA-sequencing data have been obtained in three independent experiments. A representative dataset has been deposited in GEO under accession number GSE243351. Gene ontology (GO) analysis was carried out with top 200 genes induced by ΔN-Mrtf. Gene names were converted to nearest human homolog and processed by ShinyGO (http://bioinformatics.sdstate.edu/go) (Ge et al., 2020).

For RT-qPCR, two-cell embryos were injected animally with 500 pg ΔN-Mrtf RNA and cultured until stage 9-9.5 when ectodermal explants were excised. When sibling embryos reached stage 11, the explants were harvested for total RNA extraction. cDNA was made as described above and the reactions were amplified using a CFX96 light cycler (Bio-Rad) with Universal SYBR Green Supermix (Bio-Rad). Primer sequences used for RT-qPCR are shown in Table S1. The reaction mixture consisted of 1× Power SYBR Green PCR Master Mix, 0.3 μM primers and 1 μl of cDNA in a total volume of 10 μl. The ΔΔCT method was used to quantify the results. All samples were normalized to control explants. Transcripts for *elongation factor 1a1* (*ef1a1*) were used for normalization. Data are representative of two to three independent experiments and are shown as means±s.d.

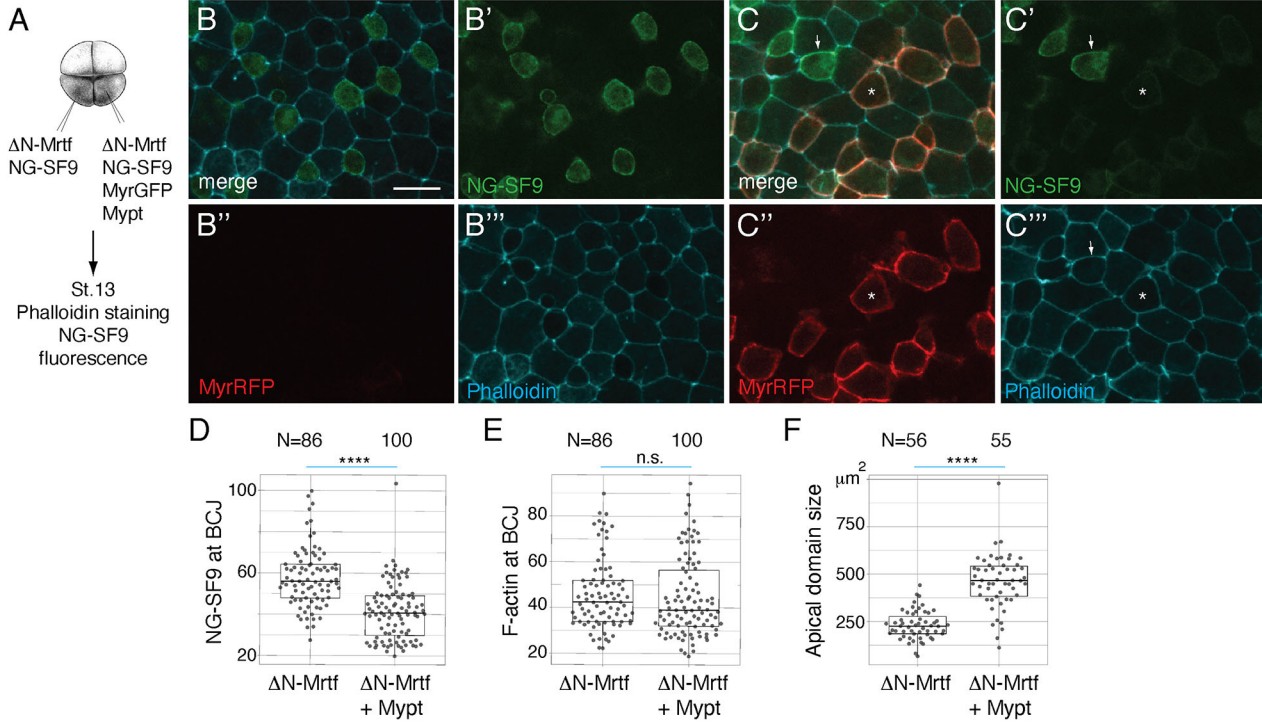

**Fig. 7. Myosin phosphatase reduces Myosin II and increases apical domain size in cells expressing ΔN-Mrtf.** (A) Scheme of injections. Four-cell embryos were injected into one ventral-animal blastomere with 25 pg mNeonGreen-SF9 (NG-SF9), 250 pg ΔN-Mrtf, 20 pg MyrRFP and 100 pg Mypt RNAs, as indicated. Injected embryos were fixed at stage 13, stained with phalloidin and imaged. (B-C‴) Images of cells mosaically expressing ΔN-Mrtf and NG-SF9 alone (B-B‴) or with Mypt and MyrRFP (C-C‴). SF9 fluorescence in ΔN-Mrtf-expressing cells becomes weaker upon co-expression of Mypt labelled with MyrRFP. (C-C‴) Stars mark cells expressing Mypt, ΔN-Mrtf, NG-SF9 and MyrRFP. Arrows mark cells expressing only ΔN-Mrtf and NG-SF9. (D-F) Quantification of NG-SF9 intensity (D), phalloidin intensity (E) or apical domain size (F) in cells expressing ΔN-Mrtf and NG-SF9 alone (ΔN-Mrtf) or with MyrRFP and Mypt (ΔN-Mrtf+Mypt). Intensities of NG-SF9 and F-actin at bicellular junctions (BCJs) were measured at the junction between a NG-SF9-expressing cell and a non-NG-SF9-expressing cell. N=numbers of cells (F) or junctions (D,E) measured. Scale bar: 20 μm. Mann–Whitney test: ****$P<0.0001$; n.s., not significant. The box plots indicate the median and quartiles.

## Fluorescent imaging and phalloidin staining

For analyses of apical cell domain size and fluorescence intensities, 4- to 8-cell embryos were injected with 10-50 pg Myr-GFP or Myr-RFP; 100-200 pg ΔN-Mrtf or ΔNΔC-Mrtf; 1 ng ΔC-Mrtf; 200 pg ΔB-Mrtf; 25-50 pg SF9-mNeonGreen; Tricellulin-mCherry; 100 pg MyptT696A RNAs; 40 ng CoMO or Mrtfa MO into one ventro-animal or dorso-animal blastomere. The embryos were fixed in 3.7% formaldehyde in PBS for 30 min when control embryos reached the indicated stages. Ventral ectoderm or neural plate was excised and imaged. To detect F-actin, 4- to 8-cell embryos were injected with RNAs and MO into one ventro-animal or dorso-animal blastomere, and the injected embryos were fixed in 3.7% formaldehyde in PBS for 30 min or 1 h when control embryos reached stage 13 or 16-17. After washing with PBS, embryos were incubated with Alexa-568-phalloidin or Alexa-647-phalloidin in PBS-Tween overnight at 4°C or 1 h at room temperature, and ectodermal explants or neural plates were excised for imaging.

Fluorescence imaging was carried out using AxioImager microscope (Zeiss) with the Apotome attachment or BC43 spinning disc confocal microscope (Andor, Oxford Instruments). Alternatively, fluorescence imaging with the LSM880 confocal microscope (Zeiss) at MSSM Core facility was carried out.

## Time lapse imaging of *Xenopus* embryos

For time-lapse imaging, 4- to 8-cell embryos were injected with 250 pg ΔN-Mrtf or ΔC-Mrtf RNA (or 40 ng of Mrtfa MO) with 50 pg Myr-GFP or 50 pg Scarlet-UtrCH, and mNeonGreen-Sf9 RNA into one ventro-animal or dorso-animal blastomere, and cultured until stage 10.25, 11 or 13. The injected embryos were mounted in 1% low melting agarose (Lonza) on a slide coverslip attached to a silicone isolator (Grace Bio-labs) or on a glass-bottomed dish (Cellvis). Time-lapse imaging was carried out using the

AxioZoom V16 fluorescence stereomicroscope (Zeiss) equipped with the AxioCam 506 mono CCD camera (Zeiss), the LSM880 confocal microscope (Zeiss) or BC43 spinning disc confocal microscope (Andor, Oxford Instruments). Images were taken every 5 or 8.5 min over a period of 1.5 or 3.5 h. Apically constricting cells were defined as those in which the apical domain size was reduced by more than 30%.

## Cryo-sectioning and immunostaining

For cryo-sectioning, embryos were devitellinized at stage 13, fixed in 3.7% formaldehyde/PBS or Dent's fixative for 1-2 h and embedded in 15% fish gelatin/15% sucrose solution (Ossipova and Sokol, 2021). The embedded embryos were frozen in dry ice and sectioned at 10 μm using Leica CM3050 cryostat. Cryo-sections were stained with anti-GFP (Santa Cruz, 1/200) and anti-β-Catenin (Sigma, 1/200) antibodies, and Alexa488-conjugated (Invitrogen, 1/200) or Cy3-conjugated (Jackson ImmunoResearch, 1/200) secondary antibodies. Fluorescence of the stained cryosections was imaged using AxioImager fluorescent microscope (Zeiss) with the Apotome attachment.

## Quantification and statistical analyses

To quantify apical domain size, explants expressing membrane marker MyrGFP or MyrRFP, and Sf9-mNeonGreen, and those stained with Phalloidin-568 or Phalloidin-647 were imaged. After image acquisition, the apical domains of individual cells were manually outlined using a free-hand line tool in ImageJ software based on membrane fluorescence. Only mosaically expressing cells of ectodermal explants touching less than three neighbors were used for quantification. For neural plate explants, apical domain size of cells in the neural plate around hindbrain region were measured.

To quantify intensities of phalloidin, Tricellulin-mCherry (TricCherry) or SF9-mNeonGreen at the tricellular junctions, bicellular junctions or medial

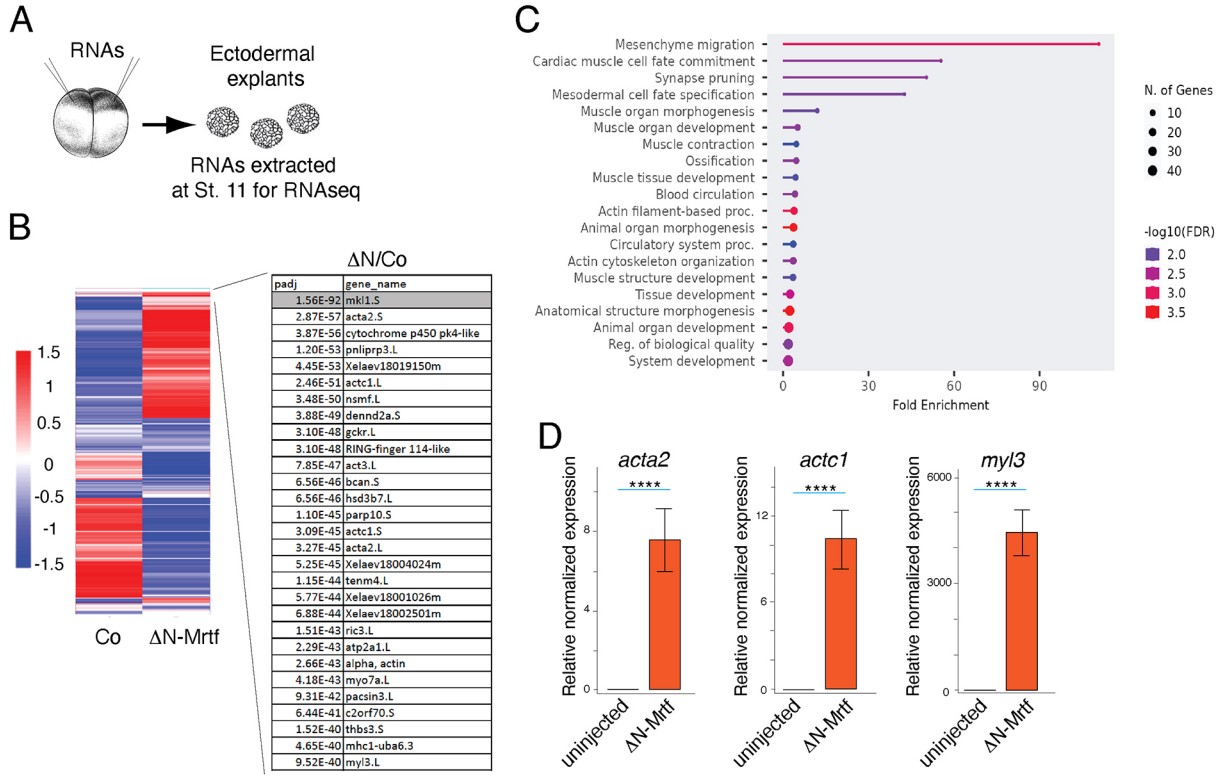

**Fig. 8. Transcriptome analysis identifies putative Mrtf target genes.** (A) Experimental scheme. Two-cell embryos were injected into two animal pole regions with 500 pg of ΔN-Mrtf RNA. Ectoderm explants were excised at stage 9-9.5. Total RNA was extracted when sibling embryos reached stage 11 for sequencing. (B) Heatmaps show relative expression of differentially expressed genes in the control and ΔN-Mrtf-expressing ectoderm. Top enriched genes are shown on the right, with adjusted *P*-value (Padj) indicated. (C) GO (gene ontology, biological processes) analysis of top 200 genes induced by ΔN-Mrtf. (D) Validation of *acta2*, *actc1* and *myl3* regulation by ΔN-Mrtf using quantitative RT-PCR. Mann–Whitney test: ****$P<0.0001$.

apical cortex, explants expressing MyrGFP, Sf9-mNeonGreen or TricCherry were stained with Phalloidin-568 or Phalloidin-647. After image acquisition, a circle tool with the same region of interest (ROI), for tricellular junctions or with a larger ROI, for medial apical cortex and a line tool for bicellular junctions were used in ImageJ to measure intensities of SF9-mNeonGreen, Tric-Cherry or Phalloidin at different cellular locations. To quantify expression levels of SF9-mNeonGreen, whole cells were manually outlined using a free-hand line tool and SF9 intensity was measured.

To quantify length-to-width ratios in superficial ectodermal cells expressing MyrGFP in the presence or absence of ΔN-Mrtf, images of ectodermal cells were acquired after immunostaining of cryo-sectioned stage 13 neurulae. Cell length along the apical to basal axis and width at the apical surface were measured using a free-hand line tool in ImageJ to quantify length-to-width ratios of superficial ectoderm cells.

For statistical analysis, a Mann–Whitney test or a Chi-square test was used to determine statistical significance between two groups. For pairwise statistical analysis of multiple groups, a one-way ANOVA Kruskal–Wallis test or a Bonferroni multiple comparison test was used. Ns, non-significant ($P>0.05$); *$P<0.05$, **$P<0.01$, ***$P<0.001$ and ****$P<0.0001$.

### Whole-mount *in situ* hybridization

Whole-mount *in situ* hybridization was carried out as described previously (Harland, 1991). Digoxygenin-11-UTP labelled RNA probes were synthesized by *in vitro* transcription with linearized *mrtfa.S*, *srf.L* and *sox2.L* DNA templates using T3 or T7 RNA polymerase (Promega) and the RNA labeling mix containing digoxygenin-11-UTP (Roche). Wild-type pigmented *Xenopus* neurula embryos at stage 14 and 15 were used in whole-mount *in situ* hybridization and after chromogenic reaction the embryos were bleached (Mayor et al., 1995). After whole-mount *in situ* hybridization, embryos were imaged using a Leica M12 stereomicroscope.

### Luciferase activity assays and immunoblotting

For luciferase activity assays, 20 pg of *3DA-Luc* reporter plasmid DNA (Posern et al., 2002) was co-injected with RNAs into two animal blastomeres of 4- to 8-cell embryos. Luciferase activity was measured when sibling embryos reached stage 10+ to 10.5, as described previously (Itoh et al., 2014).

For immunoblot analysis, 4- to 8-cell embryos were injected with RNAs encoding Mrtf constructs into two or four animal blastomeres. Whole-cell lysates were prepared from five embryos with 85 μl of the lysis buffer (50 mM Tris-HCl at pH 7.6, 50 mM NaCl, 1 mM EDTA, 1% Triton X-100, 10 mM NaF, 1 mM $Na_3VO_4$ and 1 mM PMSF), when sibling embryos reached stage 10.5 and the lysates were collected after centrifugation for 4 min at 16,000 *g*. Protein lysates were separated by SDS-PAGE and subjected to immunoblotting as previously described (Itoh et al., 2005). The following primary and secondary antibodies were used: mouse anti-FLAG (M2, Sigma, 1/1000), rabbit anti-Erk1/2 (9102, Cell Signaling, 1/1000), mouse anti-αTubulin (Developmental Studies Hybridoma Bank, 1/4000), and HRP-conjugated anti-rabbit and anti-mouse antibodies (Jackson ImmunoResearch, 1/2000). The detection was carried out by enhanced chemiluminescence as described previously (Itoh et al., 2005), using the ChemiDoc MP Imager (BioRad).

### Acknowledgements

We thank Ron Prywes, Guido Posern, Lance Davidson, Jim Smith and Ann Miller for plasmids; Mark Corkins and Aurelian Radu for advice on RNA sequence analysis; Miho Matsuda for comments on the manuscript; and members of the Sokol laboratory for discussions. We are especially grateful to Miho Matsuda for help with the statistical analysis and time-lapse imaging of gastrula ectoderm and neuroectoderm (Movies 1 and 2). We acknowledge the help and advice from the ISMMS Microscopy Core facility and the use of the LSM880 confocal microscope.

### Competing interests

The authors declare no competing or financial interests.

## Author contributions
Conceptualization: K.I., O.O., S.Y.S.; Data curation: K.I., O.O., S.Y.S.; Formal analysis: K.I., O.O.; Funding acquisition: S.Y.S.; Investigation: K.I., O.O.; Methodology: K.I., O.O., S.Y.S.; Project administration: S.Y.S.; Supervision: S.Y.S.; Validation: K.I., O.O., S.Y.S.; Visualization: K.I., O.O.; Writing – original draft: K.I., O.O., S.Y.S.; Writing – review & editing: K.I., O.O., S.Y.S.

## Funding
This study was supported by the National Institute of General Medical Sciences (R35GM122492 to S.Y.S.). Open Access funding provided by the Icahn School of Medicine at Mount Sinai. Deposited in PMC for immediate release.

## Data and resource availability
A representative dataset for RNA sequencing has been deposited in GEO under accession number GSE243351. Other relevant data and details of resources can be found within the article and its supplementary information.

## The people behind the papers
This article has an associated 'The people behind the papers' interview with some of the authors.

## Peer review history
The peer review history is available online at https://journals.biologists.com/dev/lookup/doi/10.1242/dev.204681.reviewer-comments.pdf

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
