## [Peer Review File · Development (Cambridge, England)]

Myocardin-related transcription factor regulates actomyosin contractility and apical junction remodeling during vertebrate neural tube closure

Keiji Itoh, Olga Ossipova and Sergei Y. Sokol
DOI: 10.1242/dev.204681

Editor: James M. Wells

Review timeline

Original submission:	27 September 2023
Editorial decision:	21 November 2023
Resubmission:	29 January 2025
Editorial decision:	10 March 2025
First revision:	30 June 2025
Editorial decision:	22 July 2025
Second revision:	23 July 2025
Accepted:	24 July 2025

Original submission

First decision letter

MS ID#: DEVELOP/2023/202379

MS TITLE: Myocardin-related transcription factors regulate morphogenetic events in vertebrate embryos by controlling F-actin organization and apical constriction

AUTHORS: Keiji Itoh, Olga Ossipova, Miho Matsuda, and Sergei Y Sokol

ARTICLE TYPE: Research Article

Dear Dr. Sokol,

I have now received all the referees reports on the above manuscript, and have reached a decision. The referees' comments are appended below, or you can access them online: please go to BenchPress and click on the 'Manuscripts with Decisions' queue in the Author Area.

As you will see from their reports, the referees recognise the potential of your work, but they also raise significant concerns about it. In particular, two referees felt that the work presented, while interesting, was still at early stages. Given the nature of these concerns, I am afraid I have little choice other than to reject the paper at this stage.

However, having evaluated the paper, I do recognise the potential importance of this work. I would therefore be prepared to consider as a new submission an extension of this study that contains new experiments, data and discussions and that address fully the major concerns of the referees. The work required goes beyond a standard revision of the paper. Please bear in mind that the referees (who may be different from the present reviewers) will assess the novelty of your work in the context of all previous publications, including those published between now and the time of resubmission.

Reviewer 1 Advance Summary and Potential Significance to Field:

The manuscript by Itoh et al. address the function of Mrft in the development of *Xenopus* embryos. Mrfts are transcription factors that have been previously been linked to actin function and to morphogenic changes. Here the authors show that similar to other systems the transactional activity as assayed by luciferase reporter was strongest when the G-actin binding domain was removed (DeltaN) and was negatively regulated by removal of the dimerization and TAD domains (deltaC). Overexpression of either construct or the full length Mrft leads to defects in both gastrulation and neural tube closure. These defects were similarly observed in depletion studies. The authors performed RNAseq to identify actin subunits and actin regulatory genes that are downstream of Mrft. They go on to show that actin levels are influenced by the Delta C and Delta N constructs. Finally they propose a model where Mrft is important for apical constriction based on cell size measurements and side projections from cryo sections.

Reviewer 1 Comments for the Author:

There is some interesting developmental phenotypes and some beautiful cell biological observations. Unfortunately the two don't combine into a coherent study and I feel that the paper in its current form is premature. To have the same developmental defect with an increase or decrease in Mrft function while not having similar cell biological defects is problematic. While I believe the results there is a lack of mechanistic insight into why this seeming discrepancy is occurring. Furthermore, I think there is a fair amount of inconsistency to the analysis. They perform both the Delta N and the Delta C for some experiments but not others. I think the mosaic analysis is quite beautiful but it does not appear to be uniformly applied to the studies. One might predict very different outcomes for tissues with isolated cells versus uniform expression of these constructs (or at the boundary) and I don't feel that the current analysis distinguishes these differences enough.

Comments:

It would be nice to see a GO analysis to validate that comment that the data in Figure S1B is primarily actin associated proteins.

Figure 3. I don't think the Cyto quantification (3C'') is a good description. I suspect this is primarily quantifying non junctional actin that is apical membrane associated. I would urge a different description. Also, while likely negative data, for comparison and continuity the Cyto should be similarly scored in all conditions.

I think it is important to provide better mosaic images of the tricellulin result (S4) and to provide quantification, which could potentially be added to Figure 3 for a deeper analysis.

The results from Figure 4 are hard to interpret in the current form. I appreciate that the authors attempted to find mostly isolated cells by restricting their qualifications with no more than 2 GFP positive neighbors. I am not sure this is enough. I suspect purely isolated cells would have a different phenotype than ones touching even two GFP positive neighbors. I think this analysis needs to be broken down into isolated, 2 cell touching and uniform GFP. It is also important to stain these lifeactGFP embryos with phalloidin so that one can observe the difference between the DeltaN-Mrft cells and their WT neighbors (since the pictures above in 3 don't look different in size). While the convex analysis might be expected to be different in isolated cells versus uniform expressing cells, I am not sure the same would be predicted for the apical cell size. Similarly, the DeltaC construct should be added to Figure 4.

It would be nice to see the Bcatenin staining in Figure 6C similar to 6B.

Minor:

In the last paragraph of the introduction apical constriction is misspelled apical construction.

Figure 3 is mislabeled in the text (e.g. Delta N-Mrft data is labeled 3C not 3D)

******* Reviewer 2 Advance Summary and Potential Significance to Field:**

In their submitted paper, Itoh et al use *Xenopus* embryos to demonstrate a requirement for Mrft function in gastrulation (blastopore closure) and neurulation (neural plate folding). Interestingly, while they show that knockdown by morpholino, and expression of constitutively active and

dominant interfering constructs all cause similar gross morphological phenotypes, the effects on F-actin distribution and cell behaviour are different. Inhibitory Mrtf constructs suppressed Mrtf-dependent transcription, reduced overall F-actin levels and inhibited apical constriction during gastrulation and neurulation. By contrast, constitutively active Mrtf led to decreased F-actin specifically at tricellular junctions and increased apical constriction in superficial ectoderm. The data presented in this paper are of a high quality and increase our understanding of Mrtf in vertebrate development. However, the cellular mechanism underlying Mrtf function in apical constriction remains unclear. I think there are some relatively simple additional experiments which would provide greater mechanistic understanding and strengthen the paper for publication in Development.

Reviewer 2 Comments for the Author:
Essential revisions:

1. Morpholino experiments: Uninjected embryos are an unsatisfactory control for morpholino experiments, as any morpholino can delay development in a non-specific manner. To control for this, the authors should compare to embryos injected with the same concentration of a standard control MO (e.g., the Gene Tools standard control that targets a human beta-globin intron mutation that causes beta-thalassemia). In addition, a rescue experiment with co-injection of the wild-type Mrtf construct should also be performed to demonstrate that the morphological defects seen are a direct result of Mrtf knockdown rather than a non-specific effect. It would also be informative to determine whether or not the ΔC -Mrtf or ΔN -Mrtf constructs can rescue the knockdown defects.

2. Mechanism for ΔN -Mrtf phenotype: It is potentially counter-intuitive that increasing activity of Mrtf would lead to a decrease in F-actin specifically at TCJs and that this in turn would lead to increased apical constriction of the expressing cells. The authors should explore this mechanism in more detail:

a. How is the localisation of other actin/actomyosin regulators affected by ΔN -Mrtf expression? For example, have the authors looked at the localisation of phospho-myosin light chain? It might be expected that this would be increased along the cell edges to give increased constriction.

b. Can the authors rule out that in the mosaic expression of ΔN -Mrtf, rather than the expressing cells increasing their apical constriction, the wild type cells are instead pushing out the expressing cells from the apical surface? From the Lifeact movie, it seems to be expressing cells that are surrounded by wild-type cells that are more prone to apical constriction, is this the case? If so, it suggests that the increased constriction is due to a mismatch in contractility between the wild type and ΔN -Mrtf cells.

c. To further address the questions in b, an additional method to determine cell contractility would be highly informative. For example, the use of small laser cuts along the cell edges of wild type and ΔN -Mrtf expressing cells in the mosaic embryos would demonstrate which cells have the greater contractility.

3. Statistics: More information should be given to justify the choice of parametric tests for statistical analysis - were data sets first tested for normality and, if so, which normality test was used? Also, for comparisons where 3 or more groups are compared (e.g., 5F, 5H, S5F, S6H) repeated student t-tests are not appropriate and an ANOVA should be used instead (if normally distributed). Moreover, there are a number of conclusions drawn from comparisons with no statistical support (e.g., 1 B&C, 2M, 5G, S2I), which should be rectified.

**** Reviewer 3 Advance Summary and Potential Significance to Field:

This paper aims to investigate the role of the MRTFs (co-factors of SRF) in early *Xenopus* development. The authors use a combination of overexpression of WT and mutant MRTFs as well as MRTF morpholinos, investigating the effect on gross morphology and on cell junction remodeling and cell shape. They show that inhibition or promotion of MRFT activity inhibits blastopore closure and neural tube closure and conclude that suppressing Mrtf activity reduced overall F-actin levels and inhibited apical constriction during gastrulation and neurulation. Promotion of Mrtf activity decreased apical constriction.

Reviewer 3 Comments for the Author:

I find in general that the paper is very descriptive and lacks mechanistic insights. Moreover, the authors initially describe phenotypes associated with expression of Mrtf mutants, in blastopore and

neural tube closure. However, all subsequent analysis of the mutant constructs is carried out on more lateral epidermal regions rather than on blastopore or neural tube folds. It is therefore currently unclear how these phenotypes relate to the cells at the blastopore or neural folds where they are undergoing dramatic morphogenetic changes. Is the regulation of apical constriction the same in these regions and at these different times of development and, if so, is this the major defect driving these phenotypes?

Specific points

1. On page 6, Figure 1C: what is the explanation for the ΔB dominant negative activity given that it shouldn't be able to bind SRF and/or lack NLS.
2. On page 9, in reference to Fig. 4A-D the authors compare the concavity of cells expressing ΔN -Mrtf to neighbouring control cells. However, the neighbouring cells are not labelled so it's not possible to perform such a comparison. Is this data missing or do the authors mean to compare with the control injected embryos in Fig. 4B?
3. Comparing the results of Fig. 3 and Fig. 4, the convex shape of the cells expressing ΔN -Mrtf is more pronounced in the later staged embryos in Fig. 4. Is this due to the staging of the embryos or due to the mixing of ΔN -Mrtf with WT cells and, therefore, differential cortical tension that promotes the shape change? This latter possibility is in the line with the image shown in Fig. 4C where it appears that clusters of ΔN -Mrtf cells appear less convex. Is this representative?
4. Data provided in Fig. 5 and 6 show that mutant constructs of Mrtf can affect apical constriction. These data should be complemented with morpholino knockdown (which the authors have at hand) to strengthen the claim that endogenous Mrtf is key to promoting this activity in vivo. In addition, what is the explanation that ΔC -Mrtf does not have the effect of making the cells more convex as ΔN -Mrtf does, even though the gross effects on blastopore closure are similar for these two constructs.
5. Fig. 5H shows that the expression of $\Delta N\Delta C$ -Mrtf increases apical size, although this result is not obvious from the Fig. 5E. How do the authors explain this phenotype? Is this a stronger dominant negative effect than ΔC alone? How does this construct affect transcription of actin-related genes and F-actin levels?
6. There are multiple instances of (data not shown). Kindly provide the data or do not reference.

Minor points

1. Fig S2I - define small CG

Resubmission

Author response to reviewers

Reviewer 1 Advance Summary and Potential Significance to Field:

The manuscript by Itoh et al. address the function of Mrtf in the development of *Xenopus* embryos. Mrfts are transcription factors that have been previously been linked to actin function and to morphogenic changes. Here the authors show that similar to other systems the transcriptional activity as assayed by luciferase reporter was strongest when the G-actin binding domain was removed (ΔN) and was negatively regulated by removal of the dimerization and TAD domains (ΔC). Overexpression of either construct or the full length Mrtf leads to defects in both gastrulation and neural tube closure. These defects were similarly observed in depletion studies. The authors performed RNAseq to identify actin subunits and actin regulatory genes that are downstream of Mrtf. They go on to show that actin levels are influenced by the ΔC and ΔN constructs. Finally they propose a model where Mrtf is important for apical constriction based on cell size measurements and side projections from cryo sections.

Reviewer 1 Comments for the Author:

There is some interesting developmental phenotypes and some beautiful cell biological observations. Unfortunately the two don't combine into a coherent study and I feel that the paper in its current form is premature. To have the same developmental defect with an increase or decrease in Mrtf function while not having similar cell biological defects is problematic. While I believe the results there is a lack of mechanistic insight into why this seeming discrepancy is

occurring. Furthermore, I think there is a fair amount of inconsistency to the analysis. They perform both the Delta N and the Delta C for some experiments but not others. I think the mosaic analysis is quite beautiful but it does not appear to be uniformly applied to the studies. One might predict very different outcomes for tissues with isolated cells versus uniform expression of these constructs (or at the boundary) and I don't feel that the current analysis distinguishes these differences enough.

In response to the reviewer's comments, we have refocused our study on the role of *Mrtfa* in the regulation of cell morphology and actomyosin contractility in *Xenopus* ectoderm. We have expanded our knockdown studies and confirmed that *Mrtfa* is required for apical domain reduction and F-actin enrichment in the neural plate and the ventral ectoderm (Figs. 1, 2, Suppl. Movie 1). Next, we assessed *Mrtfa* gain-of-function activity in gastrula ectoderm that has low background levels of actomyosin contractility (Fig. 3). We evaluated the effects of constitutively active *Mrtf* (ΔN -*Mrtfa*) on Tricellulin, F-actin and Myosin II (Figs. 4-6). Our loss- and gain-of-function analysis reveal complementary effects. Specifically, the constitutively active *Mrtf* construct stimulated SRF-dependent transcription and reduced apical domain size, whereas the *Mrtfa* morpholino and the dominant inhibitory construct had the opposite outcomes (Fig. 2, Fig. S4). We also confirmed that ΔN -*Mrtf* cell-autonomously enriches Myosin II (Fig. 5). The phenotypic effect of *Mrtf* overexpression was inhibited by *Mypt1*, the myosin phosphatase subunit (Fig. 6). This result is consistent with the hypothesis that *Mrtfa* functions by activating Myosin II, presumably by inducing specific differentially expressed targets that we identified by RNA sequencing (Fig. S5).

Prompted by the reviewer, we compared uniformly and mosaically expressing cell populations (Fig. 4). We observe that apical constriction is more pronounced in mosaic ΔN -expressing cells (compare Fig. 4C and E). We believe that apical domain reduction is stronger in these cells in the absence of pulling forces from their neighbors as we recently described in the neural plate (Matsuda et al, 2023). We hope that these new mechanistic insights into *Mrtfa* function address the concern of the reviewer.

Comments:

It would be nice to see a GO analysis to validate that comment that the data in Figure S1B is primarily actin associated proteins.

Following the referee request, GO analysis has been added (Fig. S5C).

Figure 3. I don't think the Cyto quantification (3C") is a good description. I suspect this is primarily quantifying non junctional actin that is apical membrane associated. I would urge a different description. Also, while likely negative data, for comparison and continuity the Cyto should be similarly scored in all conditions. I think it is important to provide better mosaic images of the tricellulin result (S4) and to provide quantification, which could potentially be added to Figure 3 for a deeper analysis.

The results from Figure 4 are hard to interpret in the current form. I appreciate that the authors attempted to find mostly isolated cells by restricting their qualifications with no more than 2 GFP positive neighbors. I am not sure this is enough. I suspect purely isolated cells would have a different phenotype than ones touching even two GFP positive neighbors. I think this analysis needs to be broken down into isolated, 2 cell touching and uniform GFP. It is also important to stain these lifeactGFP embryos with phalloidin so that one can observe the difference between the ΔN -*Mrtf* cells and their WT neighbors (since the pictures above in 3 don't look different in size). While the convex analysis might be expected to be different in isolated cells versus uniform expressing cells, I am not sure the same would be predicted for the apical cell size. Similarly, the ΔC construct should be added to Figure 4.

We have addressed the above comments and documented the effects of ΔN -*Mrtf* on tricellular junctions (TCJs) by monitoring levels of Tricellulin, Myosin II heavy chain and F-actin, in both mosaics and uniformly expressing ectoderm cells (Figs. 4-6). The appropriate quantification has been provided. In addition to ΔC -*Mrtf* effects on the ectoderm shown in Fig. S4, we included new data on *Mrtf* knockdown in the neural and non-neural ectoderm (Fig. 2, Suppl. Movie 1) supporting our conclusions on the role of *Mrtf* in apical domain regulation and neural tube

closure. Since the knockdown phenotype was more robust than the overexpression of ΔC -Mrtf, most of our new experiments have been done with Mrtfa morphants. Because the manuscript has significantly changed, some questions of the reviewer are no longer applicable.

It would be nice to see the Bcatenin staining in Figure 6C similar to 6B.

We included beta-catenin staining in the revised Fig. S3.

Minor:

In the last paragraph of the introduction apical constriction is misspelled apical construction. Figure 3 is mislabeled in the text (e.g. Delta N-Mrtf data is labeled 3C not 3D)

The text has been revised and checked for typos.

Reviewer 2 Advance Summary and Potential Significance to Field:

In their submitted paper, Itoh et al use *Xenopus* embryos to demonstrate a requirement for Mrtf function in gastrulation (blastopore closure) and neurulation (neural plate folding). Interestingly, while they show that knockdown by morpholino, and expression of constitutively active and dominant interfering constructs all cause similar gross morphological phenotypes, the effects on F-actin distribution and cell behaviour are different. Inhibitory Mrtf constructs suppressed Mrtf-dependent transcription, reduced overall F-actin levels and inhibited apical constriction during gastrulation and neurulation. By contrast, constitutively active Mrtf led to decreased F-actin specifically at tricellular junctions and increased apical constriction in superficial ectoderm. The data presented in this paper are of a high quality and increase our understanding of Mrtf in vertebrate development. However, the cellular mechanism underlying Mrtf function in apical constriction remains unclear. I think there are some relatively simple additional experiments which would provide greater mechanistic understanding and strengthen the paper for publication in *Development*.

Reviewer 2 Comments for the Author:

Essential revisions:

1. Morpholino experiments: Uninjected embryos are an unsatisfactory control for morpholino experiments, as any morpholino can delay development in a non-specific manner. To control for this, the authors should compare to embryos injected with the same concentration of a standard control MO (e.g., the Gene Tools standard control that targets a human beta-globin intron mutation that causes beta-thalassemia). In addition, a rescue experiment with co-injection of the wild-type Mrtf construct should also be performed to demonstrate that the morphological defects seen are a direct result of Mrtf knockdown rather than a non-specific effect. It would also be informative to determine whether or not the ΔC -Mrtf or ΔN -Mrtf constructs can rescue the knockdown defects.

As requested by the reviewer, we included a control MO in our experiments (Figs. 1 and 2) and carried out a phenotypic rescue by wild-type Mrtfa RNA to demonstrate specificity of Mrtfa MO for neural tube morphogenesis. MrtfA RNA partially rescued neural tube defects caused by the unilateral injection of Mrtfa MO (Fig. 1). We did not carry out the rescue with ΔC -Mrtfa because of its dominant interfering activity (Fig. S4).

2. Mechanism for ΔN -Mrtf phenotype: It is potentially counter-intuitive that increasing activity of Mrtf would lead to a decrease in F-actin specifically at TCJs and that this in turn would lead to increased apical constriction of the expressing cells. The authors should explore this mechanism in more detail:

- a. How is the localisation of other actin/actomyosin regulators affected by ΔN -Mrtf expression? For example, have the authors looked at the localisation of phospho-myosin light chain? It might be expected that this would be increased along the cell edges to give increased constriction.
- b. Can the authors rule out that in the mosaic expression of ΔN -Mrtf, rather than the expressing cells increasing their apical constriction, the wild type cells are instead pushing out the expressing cells from the apical surface? From the Lifeact movie, it seems to be expressing cells that are surrounded by wild-type cells that are more prone to apical constriction, is this the

case? If so, it suggests that the increased constriction is due to a mismatch in contractility between the wild type and ΔN -Mrtf cells.

c. To further address the questions in b, an additional method to determine cell contractility would be highly informative. For example, the use of small laser cuts along the cell edges of wild type and ΔN -Mrtf expressing cells in the mosaic embryos would demonstrate which cells have the greater contractility.

- a) Prompted by the reviewer, we examined the effect of constitutively active *Mrtfa* (ΔN -Mrtfa) on the proteins enriched at the tricellular junctions: (Tricellulin), F-actin and Myosin II (Figs. 4-6). In addition to phalloidin, we used mNeonGreen-SF9, a fluorescent recombinant antibody for Myosin II heavy chain.
- b) We find that ΔN -Mrtf enriches SF9 in the cell-autonomous manner (Fig. 5, 6) and that this effect was prevented by Mypt, the myosin phosphatase (Fig. 6), indicating that *Mrtfa* functions by modulating Myosin II activity. Additionally, the comparison of mosaic and uniformly-expressing cell populations was included (Fig. 4). As reviewer suggested, we observe more efficient apical constriction in mosaic ΔN -containing cells, likely due to force balancing between the expressing and non-expressing cells. The cells expressing ΔN -Mrtfa become more round (convex), appear to progressively remodel their TCJs, with Tricellulin, F-actin and Myosin II spreading over bicellular apical junctions, leading to the reduction of apical domain size. Together, our findings suggest that *Mrtf* cell-autonomously promotes apical constriction in the superficial ectoderm, as opposed to the less parsimonious explanation that it modifies the surrounding cells that push out the expressing cell.
- c) We confirmed increased actomyosin contractility by time-lapse imaging with Utrophin actin-binding domain Utr-Rfp as live marker for F-actin and mNeonGreen-SF9 as a marker for Myosin II. Cells expressing ΔN -Mrtfa exhibited membrane-associated F-actin puncta and flashes (Suppl. Movie 2). Moreover, we observed the increased Myosin II levels in cells mosaically expressing ΔN -Mrtfa (Fig. 5), supporting the conclusions obtained by the measurement of apical domain size (Fig. 3). Laser ablations on the cells with small apical domains, as proposed by the reviewer, were technically challenging and we decided not to pursue this approach.

3. Statistics: More information should be given to justify the choice of parametric tests for statistical analysis - were data sets first tested for normality and, if so, which normality test was used? Also, for comparisons where 3 or more groups are compared (e.g., 5F, 5H, 5SF, 56H) repeated student t-tests are not appropriate and an ANOVA should be used instead (if normally distributed). Moreover, there are a number of conclusions drawn from comparisons with no statistical support (e.g., 1 B&C, 2M, 5G, 52I), which should be rectified.

As requested, the statistical support that was previously missing has been included. We corrected previous statistical analysis by using ANOVA and Chi-square tests for the figures with multiple data sets. We used the Mann-Whitney and Kruskal-Wallis nonparametric tests to determine significance when the data did not seem to be normally distributed.

Reviewer 3 Advance Summary and Potential Significance to Field:

This paper aims to investigate the role of the MRTFs (co-factors of SRF) in early *Xenopus* development. The authors use a combination of overexpression of WT and mutant MRTFs as well as MRTF morpholinos, investigating the effect on gross morphology and on cell junction remodeling and cell shape. They show that inhibition or promotion of MRTF activity inhibits blastopore closure and neural tube closure and conclude that suppressing *Mrtf* activity reduced overall F-actin levels and inhibited apical constriction during gastrulation and neurulation.

Promotion of *Mrtf* activity decreased apical constriction.

Reviewer 3 Comments for the Author:

I find in general that the paper is very descriptive and lacks mechanistic insights. Moreover, the authors initially describe phenotypes associated with expression of *Mrtf* mutants, in blastopore and neural tube closure. However, all subsequent analysis of the mutant constructs is carried out on more lateral epidermal regions rather than on blastopore or neural tube folds. It is

therefore currently unclear how these phenotypes relate to the cells at the blastopore or neural folds where they are undergoing dramatic morphogenetic changes. Is the regulation of apical constriction the same in these regions and at these different times of development and, if so, is this the major defect driving these phenotypes?

To address the reviewer's concerns, the paper has been revised to specifically focus on neural and non-neural ectoderm. The less compelling blastopore studies have been removed following the reviewer's comments. The depletion of *Mrtf* with a specific MO demonstrated the expansion of the apical domain and reduced apical junctional F-actin in the neural plate and ventral ectoderm (new Fig. 2 and Suppl. Movie 1). These defects have been partly rescued by *Mrtfa* RNA (new Fig. 2). To assess *Mrtfa* gain-of-function activity at the cellular level, we used gastrula ectoderm, because it has low background levels of contractility. Since many cells in the neuroectoderm are highly contractile during neural tube closure, it is not a good model for gain-of-function assays. Taken together, our observations support the conclusion that *Mrtfa* cell-autonomously regulates apical contractility during neural tube closure.

Specific points

1. On page 6, Figure 1C: what is the explanation for the ΔB dominant negative activity given that it shouldn't be able to bind SRF and/or lack NLS.

The dominant negative activity of a construct equivalent to ΔB has been attributed to the binding of SRF. However, the analysis of ΔB has been removed for consistency, because we did not use this construct in the rest of the manuscript.

2. On page 9, in reference to Fig. 4A-D the authors compare the concavity of cells expressing ΔN -Mrtf to neighbouring control cells. However, the neighbouring cells are not labelled so it's not possible to perform such a comparison. Is this data missing or do the authors mean to compare with the control injected embryos in Fig. 4B?

The figure from the original manuscript was replaced with new data that confirm our original observations. We clarify that the convex shape of ΔN -Mrtfa-expressing cells is compared with the shape of the labeled cells lacking ΔN -Mrtfa in control embryos (new Figs. 3-5).

3. Comparing the results of Fig. 3 and Fig. 4, the convex shape of the cells expressing ΔN -Mrtf is more pronounced in the later staged embryos in Fig. 4. Is this due to the staging of the embryos or due to the mixing of ΔN -Mrtf with WT cells and, therefore, differential cortical tension that promotes the shape change? This latter possibility is in the line with the image shown in Fig. 4C where it appears that clusters of ΔN -Mrtf cells appear less convex. Is this representative?

Thank you for bringing up this interesting point. Indeed, the convex shape of ΔN -Mrtfa-expressing cells that are in uniformly expressing cell clusters is less pronounced than the mosaic ΔN -Mrtfa-expressing cells (see new Fig. 4, 5). We point out in Results and Discussion that this outcome probably reflects differential cortical tension (Winklbauer, 2015, Maitre et al. 2012, Sedzinski et al., 2016) due to the mixing of the expressing and non-expressing cells (Matsuda et al., 2023). We also note that the apical area becomes progressively more convex with the advancement of the apical constriction at later developmental stages.

4. Data provided in Fig. 5 and 6 show that mutant constructs of *Mrtf* can affect apical constriction. These data should be complemented with morpholino knockdown (which the authors have at hand) to strengthen the claim that endogenous *Mrtf* is key to promoting this activity in vivo. In addition, what is the explanation that ΔC -Mrtf does not have the effect of making the cells more convex as ΔN -Mrtf does, even though the gross effects on blastopore closure are similar for these two constructs.

The revised manuscript includes the requested effects of *Mrtf* knockdown on the neuroectoderm (Fig. 2). *Mrtfa* MO injection increased apical domain size to inhibit apical constriction in the neural plate and promoted the loss of cortical F-actin at the apical junctions in both neural and non-neural ectoderm (Fig. 2, Suppl. Movie 1). Similarly to *Mrtfa* MO, ΔC -Mrtf expands the apical

domain (Fig. S4). Conversely, ΔN -Mrtf enhanced rather than repressed F-actin at the apical junctions (Fig. 4). Thus, the cell biological mechanisms underlying gain- and loss-of- function effects of Mrtfa are very distinct, despite seemingly similar developmental phenotypes. We acknowledge that this conclusion was not clearly stated in the first version of the manuscript.

The effects on cell morphology tightly correlated with the ability of Mrtfa to induce transcription. ΔN -Mrtf transcriptional activity is required to induce convex cell shape and apical constriction (Fig. 3, Fig. S2), whereas ΔC -Mrtf lacks the transcription activation domain and has a dominant negative effect (Figs. S2, S4).

5. Fig. 5H shows that the expression of $\Delta N\Delta C$ -Mrtf increases apical size, although this result is not obvious from the Fig. 5E. How do the authors explain this phenotype? Is this a stronger dominant negative effect than ΔC alone? How does this construct affect transcription of actin- related genes and F-actin levels?

In principle, $\Delta N\Delta C$ -Mrtf does not have the C-terminal transcription activation domain and could be a dominant inhibitor of Mrtf (as suggested by the reviewer for the original figure). Prompted by the reviewer, we tested this possibility, but our additional experiments did not confirm the proposed dominant negative effect of $\Delta N\Delta C$ -Mrtf (new Fig. 3).

6. There are multiple instances of (data not shown). Kindly provide the data or do not reference. We removed our references to 'data not shown.'

Minor points

1. Fig S2I - define small CG

These data are no longer in the manuscript.

Second decision letter

MS ID#: dev.204681

MS Title: Myocardin-related transcription factor regulates actomyosin contractility and apical junction remodeling during vertebrate neural tube closure

Authors: Keiji Itoh, Olga Ossipova and Sergei Y. Sokol

Dear Dr Sokol,

I have now received all the referees' reports on the above manuscript, and have reached a decision. The referees' comments are appended below, or you can access them online: please go to:

As you will see, the referees express considerable interest in your work, but have some significant criticisms and recommend a substantial revision of your manuscript before we can consider publication. If you are able to revise the manuscript along the lines suggested, which may involve further experiments, I will be happy to receive a revised version of the manuscript. Your revised paper will be re-reviewed by one or more of the original referees, and acceptance of your manuscript will depend on your addressing satisfactorily the reviewers' major concerns. Please also note that Development will normally permit only one round of major revision. If it would be helpful, you are welcome to contact us to discuss your revision in greater detail. Please send us a point-by-point response indicating your plans for addressing the referees' comments, and we will look over this and provide further guidance.

Please attend to all of the reviewers' comments and ensure that you clearly highlight all changes made in the revised manuscript. Please avoid using 'Tracked changes' in Word files as these are lost in PDF conversion. I should be grateful if you would also provide a point-by-point response detailing

how you have dealt with the points raised by the reviewers in the 'Response to Reviewers' box. If you do not agree with any of their criticisms or suggestions please explain clearly why this is so.

Reviewer 1

Advance summary and potential significance to field

The manuscript entitled "Myocardin-related transcription factor regulates actomyosin contractility and apical junction remodeling during vertebrate neural tube closure" investigates the mechanism of action of Mrtfa in apical constriction of ectodermal cells during *Xenopus laevis* embryonic development. The study shows through loss and gain of function experiments that Mrtfa is required for neural plate folding and apical constriction of ectodermal cells. This investigation shows that actin dynamics are altered when Mrtfa is knocked down or overexpressed and transcriptomic assays indicate dysregulated expression of cytoskeletal protein regulation. The study shows that FGF signaling may regulate Mrtfa nuclear translocation in ectodermal cells and concludes that the regulation of actomyosin contractility is an essential cellular response to Mrtfa-dependent transcriptional activation.

This study identifies a novel mechanism operating in vivo for the regulation of apical constriction during neural tube formation and contributes with important knowledge to the understanding of this complex and crucial developmental process. Experiments are well designed, and overall the data is convincing and clearly presented. These discoveries will be of great interest to the broad readership of *Development* because it unveils novel mechanisms operating in vivo for the regulation of cell shape and behavior during developmental morphogenesis.

Comments for the author

Specific issues:

- 1) "We conclude that Mrtfa is necessary for apical constriction and junctional dynamics during neural tube closure, and it might function by regulating actin transcription and polymerization." Further experiments are needed to conclude on changes in actin transcription.
- 2) Similarly, the conclusion that "These results suggest that transcriptional activation is crucial for the induction of apical constriction by Mrtf." Needs to be further validated.
- 3) Discussion on why a factor that promotes ectopic apical constriction when overexpressed doesn't lead to hyperpigmentation, in contrast to other factors, is missing and could clarify the model through which this molecule acts, in contrast to other known pathways.
- 4) The supplementary movie 2 legend and text are not sufficient to understand what the movie shows in terms of which are the affected and control cells.
- 5) Assessment of level of expression of NGSF9 in control and experimental cells is needed to conclude on differential localization to BCJs.
- 6) Moreover, the difference in BCJs in mosaic samples often finds cell-cell borders shared by a control and affected cell, where it is not possible to determine which border belongs to each of the cells. Hence it is intriguing that in global/non-mosaic samples this differential enrichment of sf9 is not observed. Instead of a "cell-autonomous" phenomenon, this result could be interpreted as either an artifact of the mosaicism (ie, uneven surface of the tissue when cells have different shapes) or that the level of expression in affected cells is lower (darker overall labeling of cells) and that the border with the control cells by contrast seems brighter.
- 7) "The difference in protein localization was most pronounced in mosaically-expressing cells (compare Figs. 4C and 4E)." This is not apparent either in the examples or the graphs presented in Fig. 4.

8) "Note that Sf9 fluorescence changed most in the mosaics, indicating that the ΔN -Mrtf effects are cell-autonomous." First, to assess this difference the data should be included to compare both approaches. Second, I don't think this is the only or most appropriate explanation to the difference in Sf9 phenotype in ΔN -Mrtfa mosaically and globally misexpressed samples. The fact that the phenotype is milder in the latter suggest that there is some compensatory mechanism coming from the adjacent affected cells or from a globally affected embryo that has more chances to compensate for the perturbations to the cellular phenotype from the overexpression. Alternatively, it could be that adjacent wild-type/control cells contribute to the phenotype triggered in Mrtfa misexpressing cells, thus making the effect non-cell autonomous because it is influenced by cells surrounding those affected.

9) Although the transcriptomic approach in GOF samples is interesting, the analysis of LOF cells could potentially reveal more physiologically relevant targets of Mrtfa in vivo during neural tube closure.

10) Although the subcellular localization of Mrtfa at the onset of gastrulation is interesting, assessing it in neural plate stages when the Mrtfa knockdown was shown to affect neural tube closure will better interconnect the findings presented in this study.

11) In the discussion it is said: "we observe cell-autonomous enrichment of the Myosin II heavy chain that is evident by the accumulation of SF9 fluorescence in the cortex of ΔN -Mrtfa-containing ectoderm cells" but cortical accumulation is not that apparent, as shown in Fig. 5B. If anything, cortical fluorescence seems lower in of ΔN -Mrtfa-containing ectoderm cells. In addition, what is a little troubling is that there seems to be a discrepancy in the phenotype of NGSF9 subcellular distribution and comparison with non-perturbed cells in figures 5B and 6B. Otherwise, more clarification on what these panels in the 2 figures are showing and/or better selection of exemplar images are needed.

Reviewer 2

Advance summary and potential significance to field

Itoh et al. identified the role of the transcription factor, Mrtfa, in frog embryogenesis, particularly in neural tube closure. Mrtfa induces apical constriction via the tricellular junction formation and up-regulation of actomyosin mRNA. Since the transcriptional activity of Mrtfa is controlled by its nuclear localization, they also showed that FGF8 signaling can facilitate the nuclear localization of Mrtfa. These new findings may provide the fascinating function of Mrtfa during Xenopus neurulation in this field. But some concerns are listed below.

Comments for the author

1) The expression patterns of Mrtfa during frog neurulation is not described. Both in situ mRNA and protein expression of Mrtfa, together with expression of its cofactor, serum response factor, would be necessary to know the function of Mrtfa in neurulation. In addition, if the authors can show nuclear translocation of Mrtfa is evident during neurulation, it would be helpful to understand the role of nuclear localized Mrtfa in neural tube closure and signaling input. When a good specific antibody is not available, they can also use a GFP-fused Mrtfa protein expression for subcellular localization analysis of Mrtfa during neural tube closure.

2) Several previous reports identify the nuclear function of Mrtfa as being involved in the transcriptional activation of transcriptional targets, while the cytoplasmic localized protein appears to be unclear. Does cytoplasmic-localized Mrtfa possess no biological activity, just null or specific other activities?

3) The authors show that chordin does not facilitate nuclear localization of Mrtfa, but FGF8 does. However, the biological significance of the involvement of FGF8 in the nuclear translocation of Mrtfa remains unclear. When and how does FGF8 mediate neural tube closure in Xenopus embryos? Can FGF8 antagonists or analogs affect neural tube closure in Xenopus? If a specific role of FGF8 in neurulation was not provided, given that with the exception of Chordin, several agonists or

antagonists of BMP, Shh, and Wnt are also expressed during neurulation, the authors should have better tested effects of these secretory molecules in terms of nuclear translocation of Mrtfa.

4) The roles of apical constriction and tricellular junction formation in neurulation are not clearly shown in the manuscript. Are these two phenomena required for proper neural tube closure in *Xenopus*?

Reviewer 3

Advance summary and potential significance to field

The manuscript by Itoh et al shows that myocardin-related transcription factor (Mrtf) regulates apical contractility in *Xenopus* neural and non-neural ectoderm cells. They show that the transcriptional activity of Mrtf is required for inducing apical contractility. This is an interesting finding with potential implications for deeper understanding of neural tube morphogenesis, although the relevance to neural tube development could be further strengthened. Overall, the manuscript is well-written and clear. I recommend addressing the following issues before publication:

Comments for the author

1. The main issue is that most of the experiments that were performed in the study are focused on non-neural ectoderm cells. While the authors provide a justification for this (the contractility of neuroepithelial cells is already high), I think it is important to demonstrate whether gain of function of Mrtf has a phenotype in the neural tube itself. This will allow the authors to strengthen the relevance of their findings to the neural tube.
2. Related to the above point, to demonstrate the relevance to the neural tube, the authors should show the localization of the GFP-Mrtf construct (that was used in Fig. 7) in the neural tube at st 17-18. It would be interesting to see if there is any rostro-caudal or medio-lateral gradient of nuclear localization, possibly correlating with Fgf activity or with tensile forces.
3. Fig. 4 F, G Quantification is shown for uniform and mosaic conditions, presumably meaning that the data from different experiments has been pooled. However, the panels shown in A and C indicate that the absolute F-Actin fluorescence might differ substantially between uniform and mosaic conditions. In fact, it is difficult to see a difference in the F-actin staining between control and deltaN-Mrtf staining in the mosaic condition. The authors should comment on that and also include separate quantifications of the uniform and mosaic experiments, with consideration of different types of junctions (wt vs mut homo or heterotypic) in the mosaic condition.
4. In Fig. 5 - what kind of BCJ were quantified? Are BCJ between wildtype cells included in the mutant mosaics? This needs to be explained.
5. Mrtf is a transcription factor that binds G-actin and the authors show that actin binding reduces the transcriptional activity of Mrtf and expands cell area. This is an interesting regulatory feature of Mrtf-s, the authors could address experimentally or at least comment on whether and how it could be relevant for neural tube morphogenesis.

Second revision

Author response to reviewers' comments

Reviewer 1: SUMMARY OF THE ADVANCE MADE IN THIS PAPER AND ITS POTENTIAL SIGNIFICANCE TO THE FIELD

The manuscript entitled "Myocardin-related transcription factor regulates actomyosin contractility and apical junction remodeling during vertebrate neural tube closure" investigates the mechanism of action of Mrtfa in apical constriction of ectodermal cells during *Xenopus laevis* embryonic

development. The study shows through loss and gain of function experiments that *Mrtfa* is required for neural plate folding and apical constriction of ectodermal cells. This investigation shows that actin dynamics are altered when *Mrtfa* is knocked down or overexpressed and transcriptomic assays indicate dysregulated expression of cytoskeletal protein regulation. The study shows that FGF signaling may regulate *Mrtfa* nuclear translocation in ectodermal cells and concludes that the regulation of actomyosin contractility is an essential cellular response to *Mrtfa*-dependent transcriptional activation.

This study identifies a novel mechanism operating *in vivo* for the regulation of apical constriction during neural tube formation and contributes with important knowledge to the understanding of this complex and crucial developmental process. Experiments are well designed, and overall the data is convincing and clearly presented. These discoveries will be of great interest to the broad readership of *Development* because it unveils novel mechanisms operating *in vivo* for the regulation of cell shape and behavior during developmental morphogenesis.

SUGGESTIONS TO AUTHORS

Specific issues:

1) "We conclude that *Mrtfa* is necessary for apical constriction and junctional dynamics during neural tube closure, and it might function by regulating actin transcription and polymerization". Further experiments are needed to conclude on changes in actin transcription.

We agree with the reviewer that the quoted sentence was misleading, because the relevant experiments were only presented and discussed later. The misleading phrase "*.. it might function by regulating actin transcription*" was removed from the text.

2) Similarly, the conclusion that "These results suggest that transcriptional activation is crucial for the induction of apical constriction by *Mrtf*." Needs to be further validated.

To clarify the basis for our conclusions related to transcription, revised **Fig. 1** and **Fig. 8** contain data pertinent to transcriptional regulation by *Mrtf*. We confirmed the previous conclusion that *Mrtf* functions as a transcriptional coactivator of SRF not only in cultured mammalian cells (Cen et al., 2003; Miralles et al., 2003; Olson and Nordheim, 2010; Wang et al., 2002), but also in *Xenopus* embryos using transient transcriptional assays with a MRTF-specific reporter (**Fig. 1**). We also identified putative gene targets of *Mrtfa* in the embryo (**Fig. 8**). Given the requirement of the C-terminal domain (CTD) of MRTF in transcriptional activation (Cen et al., 2003; Miralles et al., 2003; Zaromytidou et al., 2006), we show that the $\Delta N\Delta C$ construct lacking CTD lost the ability to induce apical constriction. In an independent approach, we constructed *Mrtfa* ΔB that does not interact with SRF, its DNA binding cofactor (Zaromytidou et al., 2006). We confirmed that this construct is transcriptionally inactive (**Fig. 1D**) and is unable to trigger apical constriction in gastrula ectoderm (new **Fig. S5**). Together, these results support our conclusion that *Mrtfa* effect on apical contractility is due to *Srf*-dependent transcription.

3) Discussion on why a factor that promotes ectopic apical constriction when overexpressed doesn't lead to hyperpigmentation, in contrast to other factors, is missing and could clarify the model through which this molecule acts, in contrast to other known pathways.

Whereas the novel MRTF-dependent mechanism of apical constriction is not well understood, it appears distinct from the known mechanisms used by other known apical constriction inducers. Revised Discussion states that "apical domain reduction triggered by ΔN -*Mrtf* is observed in the absence of medioapical actomyosin enrichment and hyperpigmentation as reported for apical constriction induced by *Shroom3*, *Plekhg5*, or *Lmo7* (Haigo et al., 2003; Matsuda et al., 2022; Popov et al., 2018). The hyperpigmentation probably reflects melanosome maturation and redistribution to the apical cortex (Fairbank et al., 2006). These differences point to a distinct apical constriction mechanism underlying *Mrtfa* effects".

4) The supplementary movie 2 legend and text are not sufficient to understand what the movie shows in terms of which are the affected and control cells.

We added more details to the description of the movie and point to the affected cells. Since cells undergo apical constriction at different times, only a few cells reduce their apical domains during the course of the movie as compared to their neighbors.

5) Assessment of level of expression of NG-SF9 in control and experimental cells is needed to conclude on differential localization to BCJs.

Prompted by the referee, we measured NG-SF9 intensity in whole cells and found that it was unaffected by the presence of ΔN -Mrtf (new Fig. 6I, J).

6) Moreover, the difference in BCJs in mosaic samples often finds cell-cell borders shared by a control and affected cell, where it is not possible to determine which border belongs to each of the cells. Hence it is intriguing that in global/non-mosaic samples this differential enrichment of sf9 is not observed. Instead of a "cell-autonomous" phenomenon, this result could be interpreted as either an artifact of the mosaicism (ie, uneven surface of the tissue when cells have different shapes) or that the level of expression in affected cells is lower (darker overall labeling of cells) and that the border with the control cells by contrast seems brighter.

Additional quantification confirms that NG-SF9 is enriched by ΔN -Mrtfa at BCJs in both mosaic and uniformly-expressing cells as compared to the controls (revised Fig. 6B-F). We clarify that fluorescence was measured at the junction of ΔN -Mrtf expressing and control cell for mosaic cells, or at the junction between two control cells. By contrast, we found no difference in the total NG-SF9 intensity between the cells with or without ΔN -Mrtf (Fig. 6I, J).

7) "The difference in protein localization was most pronounced in mosaically-expressing cells (compare Figs. 4C and 4E)." This is not apparent either in the examples or the graphs presented in Fig. 4.

Thank you for noticing this inconsistency. Careful quantification of Tric and F-actin intensities in "uniform" or "mosaic" cells with and without ΔN expression is shown in revised Fig. 5F-I. The text was corrected to say that "the change in Tricellulin and F-actin distribution was observed in both mosaic cells and the cells uniformly expressing ΔN -Mrtfa.

8) "Note that Sf9 fluorescence changed most in the mosaics, indicating that the ΔN -Mrtf effects are cell-autonomous." First, to assess this difference the data should be included to compare both approaches. Second, I don't think this is the only or most appropriate explanation to the difference in Sf9 phenotype in ΔN -Mrtfa mosaically and globally misexpressed samples. The fact that the phenotype is milder in the latter suggest that there is some compensatory mechanism coming from the adjacent affected cells or from a globally affected embryo that has more chances to compensate for the perturbations to the cellular phenotype from the overexpression. Alternatively, it could be that adjacent wild-type/control cells contribute to the phenotype triggered in Mrtfa misexpressing cells, thus making the effect non-cell autonomous because it is influenced by cells surrounding those affected.

The data on Sf9 fluorescence in cells uniformly expressing ΔN -Mrtf have been added to the revision (revised Fig. 6), as requested. SF9 was enriched at BCJ in both uniform and mosaic ΔN -Mrtf expressing cells. These results raise a possibility that the change in cell shape is more pronounced in mosaic cells because of the mismatched cortical tension at the junction between an ΔN -Mrtf-expressing and a control cell. The statement of the *cell-autonomous effect* has been removed and we described the alternative explanations suggested by the reviewer.

9) Although the transcriptomic approach in GOF samples is interesting, the analysis of LOF cells could potentially reveal more physiologically relevant targets of Mrtfa in vivo during neural tube closure.

We agree with the reviewer that loss-of-function experiments are needed to elucidate physiological targets of Mrtfa. However, the current study focuses on cell shape regulation by Mrtfa rather than on its gene targets. Since the proposed analysis of transcriptional targets is beyond the scope of the paper, we decided to pursue it in subsequent studies.

10) Although the subcellular localization of *Mrtfa* at the onset of gastrulation is interesting, assessing it in neural plate stages when the *Mrtfa* knockdown was shown to affect neural tube closure will better interconnect the findings presented in this study.

The analysis of the subcellular localization of *Mrtfa* is hindered in the absence of antibodies to the endogenous protein. Although we included the initial data with overexpressed protein into the first revision, we decided to remove these results because they are incomplete and much more needs to be done for proper interpretation.

11) In the discussion it is said: "we observe cell-autonomous enrichment of the Myosin II heavy chain that is evident by the accumulation of SF9 fluorescence in the cortex of ΔN -*Mrtfa*-containing ectoderm cells" but cortical accumulation is not that apparent, as shown in Fig. 5B. If anything, cortical fluorescence seems lower in ΔN -*Mrtfa*-containing ectoderm cells. In addition, what is a little troubling is that there seems to be a discrepancy in the phenotype of NGSF9 subcellular distribution and comparison with non-perturbed cells in figures 5B and 6B. Otherwise, more clarification on what these panels in the 2 figures are showing and/or better selection of exemplar images are needed.

We apologize for the misunderstanding due to the imprecise description in the text. The text has been corrected to refer to 'lateral junctions' instead of 'cortex'. We agree that the enhancement of Sf9 at the lateral junctions is accompanied by its reduction at the medioapical cortex in ΔN -*Mrtf* expressing cells. This is quantified in the revised Fig. 6E-H. Additionally, to clarify the perceived discrepancy between former Fig. 5B and 6B panels, the panels were replaced with more representative subcellular localization images for NG-SF9.

Reviewer 2: SUMMARY OF THE ADVANCE MADE IN THIS PAPER AND ITS POTENTIAL SIGNIFICANCE TO THE FIELD

Itoh et al. identified the role of the transcription factor, *Mrtfa*, in frog embryogenesis, particularly in neural tube closure. *Mrtfa* induces apical constriction via the tricellular junction formation and up-regulation of actomyosin mRNA. Since the transcriptional activity of *Mrtfa* is controlled by its nuclear localization, they also showed that FGF8 signaling can facilitate the nuclear localization of *Mrtfa*. These new findings may provide the fascinating function of *Mrtfa* during *Xenopus* neurulation in this field. But some concerns are listed below.

SUGGESTIONS TO AUTHORS

1) The expression patterns of *Mrtfa* during frog neurulation is not described. Both *in situ* mRNA and protein expression of *Mrtfa*, together with expression of its cofactor, serum response factor, would be necessary to know the function of *Mrtfa* in neurulation. In addition, if the authors can show nuclear translocation of *Mrtfa* is evident during neurulation, it would be helpful to understand the role of nuclear localized *Mrtfa* in neural tube closure and signaling input. When a good specific antibody is not available, they can also use a GFP-fused *Mrtfa* protein expression for subcellular localization analysis of *Mrtfa* during neural tube closure.

We include *in situ* hybridization data for neurula embryos at stages 14 and 15 (new Fig. S2) showing that both *Mrtfa* and *Srf* are broadly expressed in neuroectoderm. Additionally, a stronger *Srf* signal is detected in the somitic mesoderm.

Prompted by the reviewer, we also attempted to examine the localization of GFP-*Mrtf* during neurulation, but the data on the distribution of the exogenous protein were variable depending on the protein level. We felt that this analysis was premature to present in the current study.

2) Several previous reports identify the nuclear function of *Mrtfa* as being involved in the transcriptional activation of transcriptional targets, while the cytoplasmic localized protein appears to be unclear. Does cytoplasmic-localized *Mrtfa* possess no biological activity, just null or specific other activities?

Although *Mrtfa* may, in principle, have additional roles in the cytoplasm, its only known function is transcriptional control.

3) The authors show that chordin does not facilitate nuclear localization of Mrtfa, but FGF8 does. However, the biological significance of the involvement of FGF8 in the nuclear translocation of Mrtfa remains unclear. When and how does FGF8 mediate neural tube closure in *Xenopus* embryos? Can FGF8 antagonists or analogs affect neural tube closure in *Xenopus*? If a specific role of FGF8 in neurulation was not provided, given that with the exception of Chordin, several agonists or antagonists of BMP, Shh, and Wnt are also expressed during neurulation, the authors should have better tested effects of these secretory molecules in terms of nuclear translocation of Mrtfa.

The questions are definitely interesting and relevant to FGF signaling during neural tube closure but they are beyond the scope of the current manuscript. To focus the paper on the cell behavior during neurulation, we decided to withdraw the figure related to the nuclear translocation of Mrtfa from the paper. Future studies will properly examine the regulation of Mrtf nuclear localization.

4) The roles of apical constriction and tricellular junction formation in neurulation are not clearly shown in the manuscript. Are these two phenomena required for proper neural tube closure in *Xenopus*?

The critical roles of tricellular junction remodeling and apical constriction in neurulation have been described in several previous studies (Nandadasa et al., 2009, Nishimura et al., 2012, Baldwin et al., 2022, reviewed by Vijayraghavan, 2017, Suzuki, 2012). The revised discussion summarizes how these processes impact neural tube closure.

Reviewer 3: SUMMARY OF THE ADVANCE MADE IN THIS PAPER AND ITS POTENTIAL SIGNIFICANCE TO THE FIELD

The manuscript by Itoh et al shows that myocardin-related transcription factor (Mrtf) regulates apical contractility in *Xenopus* neural and non-neural ectoderm cells. They show that the transcriptional activity of Mrtf is required for inducing apical contractility. This is an interesting finding with potential implications for deeper understanding of neural tube morphogenesis, although the relevance to neural tube development could be further strengthened. Overall, the manuscript is well-written and clear. I recommend addressing the following issues before publication:

SUGGESTIONS TO AUTHORS

1. The main issue is that most of the experiments that were performed in the study are focused on non-neural ectoderm cells. While the authors provide a justification for this (the contractility of neuroepithelial cells is already high), I think it is important to demonstrate whether gain of function of Mrtf has a phenotype in the neural tube itself. This will allow the authors to strengthen the relevance of their findings to the neural tube.

The first version of the paper included the phenotype of ΔN -Mrtf overexpression with extensive neural tube defects. We removed while responding to previous critique, because it may reflect non-physiologically high transcriptional activity of Mrtf. We decided not to add it back because it does not add significantly to the understanding of the endogenous role of Mrtfa during neurulation.

2. Related to the above point, to demonstrate the relevance to the neural tube, the authors should show the localization of the GFP-Mrtf construct (that was used in Fig. 7) in the neural tube at st 17-18. It would be interesting to see if there is any rostro-caudal or medio-lateral gradient of nuclear localization, possibly correlating with Fgf activity or with tensile forces.

We attempted to examine the localization of GFP-Mrtf during neurulation, but the data on the distribution of the exogenous protein were variable depending on protein expression level. We felt that this analysis is premature and beyond the scope of the present manuscript. We therefore removed the figure with our initial Mrtfa localization data from the manuscript. Future studies need to develop an antibody to endogenous protein to properly investigate this issue.

3. Fig. 4 F, G Quantification is shown for uniform and mosaic conditions, presumably meaning that the data from different experiments has been pooled. However, the panels shown in A and C indicate that the absolute F-Actin fluorescence might differ substantially between uniform and mosaic conditions. In fact, it is difficult to see a difference in the F-actin staining between control

and deltaN-Mrtf staining in the mosaic condition. The authors should comment on that and also include separate quantifications of the uniform and mosaic experiments, with consideration of different types of junctions (wt vs mut homo or heterotypic) in the mosaic condition.

As requested, we now include additional quantification of the Tric and F-actin levels in cells with uniform or mosaic ΔN -Mrtf expression (revised Fig. 5F-I). The legend clarifies how the intensities of Tric and F-actin at TCJ and BCJ were measured at the junctions. We confirmed that ΔN -Mrtf causes the redistribution of Tric and F-actin from TCJ to BCJ in both uniformly expressing and mosaic cells.

4. In Fig. 5 - what kind of BCJ were quantified? Are BCJ between wildtype cells included in the mutant mosaics? This needs to be explained.

Revised Fig. 6 clarifies that NG-SF9 and phalloidin fluorescence intensities were measured at the junction of ΔN -Mrtf expressing and control cell (ΔN), and at the junction between the control cells (co). Total NG-SF9 intensities were equal in the absence or presence of ΔN -Mrtf. We confirm that NG-SF9 was enriched at BCJ in both mosaic and uniformly expressing ΔN -Mrtf cells (revised Fig. 6E, F).

5. Mrtf is a transcription factor that binds G-actin and the authors show that actin binding reduces the transcriptional activity of Mrtf and expands cell area. This is an interesting regulatory feature of Mrtf-s, the authors could address experimentally or at least comment on whether and how it could be relevant for neural tube morphogenesis.

In the revised discussion we note that the F-actin polymerization relieves G-actin inhibition and upregulates Mrtfa activity. Thus, Mrtfa acts in a feedback loop: F-actin polymerization that depends on Mrtf is likely to further stimulate G-actin transcription via Mrtf.

Third decision letter

MS ID#: dev.204681R1

MS Title: Myocardin-related transcription factor regulates actomyosin contractility and apical junction remodeling during vertebrate neural tube closure

Authors: Keiji Itoh, Olga Ossipova and Sergei Y. Sokol

Dear Dr Sokol,

I have now received all the referees reports on the above manuscript, and have reached a decision. The referees' comments are appended below, or you can access them online: please go to .

The overall evaluation is positive and we would like to publish a revised manuscript in Development, however one reviewer noticed some typos that you can fix and upload a revised manuscript. Once that is done I'll be happy to formally accept the manuscript.

Reviewer 1

Advance summary and potential significance to field

The manuscript entitled "Myocardin-related transcription factor regulates actomyosin contractility and apical junction remodeling during vertebrate neural tube closure" investigates the mechanism of action of Mrtfa in apical constriction of ectodermal cells during *Xenopus laevis* embryonic development. The study shows through loss and gain of function experiments that Mrtfa is required

for neural plate folding and apical constriction of ectodermal cells. This investigation shows that actin dynamics are altered when *Mrtfa* is knocked down or overexpressed and transcriptomic assays indicate dysregulated expression of cytoskeletal protein regulation. The study concludes that the regulation of actomyosin contractility is an essential cellular response to *Mrtfa*-dependent transcriptional activation.

This study identifies a novel mechanism operating *in vivo* for the regulation of apical constriction during neural tube formation and contributes with important knowledge to the understanding of this complex and crucial developmental process. Experiments are well designed, the data is convincing and clearly presented. These discoveries will be of great interest to the broad readership of *Development* because it unveils novel mechanisms operating *in vivo* for the regulation of cell shape and behavior during developmental morphogenesis.

The authors have address all the concerns noted in the original submission with new experiments, clarification of text, data analysis and presentation.

Minor issues:

- 1) Reference in the text to "Fig. 1C" needs to be changed to Fig. 1C-D to include the data on ΔB -Mrtf presented in Fig. 1D and not in Fig. 1C.
- 2) Page 10, first paragraph, 2nd line, change "N-cadeherin" by N-cadherin.

Comments for the author

The authors have address all the concerns noted in the original submission with new experiments, clarification of text, data analysis and presentation.

Minor issues:

- 1) Reference in the text to "Fig. 1C" needs to be changed to Fig. 1C-D to include the data on ΔB -Mrtf presented in Fig. 1D and not in Fig. 1C.
- 2) Page 10, first paragraph, 2nd line, change "N-cadeherin" by N-cadherin.

Reviewer 2

Advance summary and potential significance to field

Comments for the author

The authors clarified some points requested in the initial review by providing new expression data, and the revised manuscript appears improved. They have carefully revised the manuscript, making it easier for readers outside the *Xenopus* field to follow.

Reviewer 3

Advance summary and potential significance to field

Comments for the author

The authors have responded to my comments, clarified the experimental limitations and improved the manuscript.

Third Revision

Author response to reviewers' comments

Reviewer 1

Minor issues:

1) Reference in the text to "Fig. 1C" needs to be changed to Fig. 1C-D to include the data on DeltaB-Mrtf presented in Fig. 1D and not in Fig. 1C.

Done

2) Page 10, first paragraph, 2nd line, change "N-cadeherin" by N-cadherin.

Corrected

Fourth decision letter

MS ID#: dev.204681R2

MS Title: Myocardin-related transcription factor regulates actomyosin contractility and apical junction remodeling during vertebrate neural tube closure

Authors: Keiji Itoh, Olga Ossipova and Sergei Y. Sokol

Dear Dr Sokol,

I am happy to tell you that your manuscript has been accepted for publication in Development, pending our standard publication integrity checks.